# Conserved arginine residues in synaptotagmin 1 regulate fusion pore expansion through membrane contact

Sarah B. Nyenhuis[1,4], Nakul Karandikar [1], Volker Kiessling [2,3], Alex J. B. Kreutzberger [2,3,5], Anusa Thapa[1], Binyong Liang[2,3], Lukas K. Tamm [2,3] & David S. Cafiso [1,2,3✉]

Synaptotagmin 1 is a vesicle-anchored membrane protein that functions as the $Ca^{2+}$ sensor for synchronous neurotransmitter release. In this work, an arginine containing region in the second C2 domain of synaptotagmin 1 (C2B) is shown to control the expansion of the fusion pore and thereby the concentration of neurotransmitter released. This arginine apex, which is opposite the $Ca^{2+}$ binding sites, interacts with membranes or membrane reconstituted SNAREs; however, only the membrane interactions occur under the conditions in which fusion takes place. Other regions of C2B influence the fusion probability and kinetics but do not control the expansion of the fusion pore. These data indicate that the C2B domain has at least two distinct molecular roles in the fusion event, and the data are consistent with a model where the arginine apex of C2B positions the domain at the curved membrane surface of the expanding fusion pore.

---

[1] Department of Chemistry, University of Virginia, Charlottesville, VA, USA. [2] Department of Molecular Physiology and Biological Physics, University of Virginia, Charlottesville, VA, USA. [3] Center for Membrane Biology, University of Virginia, Charlottesville, VA, USA. [4] Present address: Laboratory of Cell and Molecular Biology, National Institute of Diabetes and Digestive and Kidney Diseases, NIH, Bethesda, MD, USA. [5] Present address: Department of Cell Biology, Harvard Medical School and Program in Cellular and Molecular Medicine, Boston Children's Hospital, Boston, MA, USA. ✉email: cafiso@virginia.edu

Synchronous neurotransmitter release is a highly regulated process that results from the $Ca^{2+}$-triggered fusion of synaptic vesicles with the presynaptic plasma membrane. This fusion process is driven by the assembly of the three neuronal SNAREs (soluble N-ethylmaleimide sensitive receptor proteins), syntaxin (Syx) and SNAP-25 in the plasma membrane and synaptobrevin (Syb) in the vesicle membrane[1,2]. The assembly of the SNAREs into a four helical bundle or SNARE complex is highly favorable and provides the energy required to overcome the energetic barriers to fusion. Several other proteins interact with the SNAREs and are critical to the proper assembly of the complex, including Munc18, Munc13, and complexin[3]. The $Ca^{2+}$-sensor for this process is the vesicle-anchored protein, synaptotagmin 1 (Syt1), and the molecular event that initiates fusion is the binding of $Ca^{2+}$ to the two C2 domains of Syt1[4]. However, the mechanism by which $Ca^{2+}$ binding to the Syt1 C2 domains drives fusion is not understood.

The C2 domains of Syt1 are known to interact with and penetrate the membrane interface upon $Ca^{2+}$ binding[5,6]. The second C2 (C2B) domain of Syt1 has a highly charged polybasic face that allows the domain to also associate in a $Ca^{2+}$-independent manner to negatively charged membrane interfaces[7]. This interaction is weak to monovalent lipid such as phosphatidylserine (PS) but much stronger to membrane interfaces containing the multivalent lipid phosphatidylinositol-4,5-bisphosphate (PI(4,5)P$_2$—referred to here as PIP$_2$)[8], which has a valence of about $-4$[9]. In the presence of $Ca^{2+}$, the $Ca^{2+}$ binding loops and polybasic face work cooperatively to drive membrane association of the C2B domain. Synaptotagmin 1 is also known to associate with SNAREs, and several crystal structures have been generated showing the C2 domains of Syt1 in association with assembled SNAREs[10,11]. However, the interactions are heterogeneous, and they are not observed in the presence of ATP under conditions that resemble those found within the cell[12]. Synaptotagmin 1 may trigger fusion by altering the local lipid bilayer structure, thereby stimulating a conformational change in the nearby SNARE complex catalyzing membrane fusion[13]. There is also evidence that Syt1 might alter the vesicle membrane/plasma membrane distance, and conceivably this event might trigger SNARE assembly[14,15]. Synaptotagmin 1 also de-mixes PS upon membrane association[16], and this change in charged lipid distribution might modulate the membrane association of other proteins, such as Munc13 or the juxta-membrane regions of the SNAREs.

In addition to its charged polybasic face, the C2B domain of Syt1 has several highly conserved arginine residues in a region opposite the $Ca^{2+}$-binding loops that we will refer to as the arginine apex. Mutating this arginine apex is reported to dramatically depress evoked excitatory postsynaptic currents in hippocampal neurons, and this apex is proposed to allow the C2B domain to bridge across bilayers[17]. There is also direct evidence that the arginine apex associates with lipid bilayers and that it could facilitate contact of the C2B domain with two opposing bilayers[18]. In the present work, we use spin labels incorporated into sites in the apex to demonstrate that the arginine apex is associated with the membrane when the C2B domain is membrane bound and that membrane contact is dependent upon conserved arginine residues in the domain. The arginine apex also associates with the SNAREs in a manner consistent with crystal structures. However, unlike the membrane interaction, the interaction with the SNAREs is eliminated under conditions where fusion takes place. Using a single-particle fusion assay with purified secretory granules, we find that the arginine apex does not affect the probability or kinetics of fusion but is necessary for the rapid release of secretory content, indicating that the apex has a role in controlling the expansion of the fusion pore. This result

contrasts with that for the polybasic face, which alters the kinetics and probability of fusion but does not alter the rate of fusion pore expansion. The data indicate that there are two distinct molecular roles for Syt1 where different regions of the C2B domain make different contributions to the fusion process. We propose a mechanism for the action of Syt1 where the arginine apex of C2B helps position the domain at the negatively curved membrane surface that forms during the opening and expansion of the fusion pore.

## Results

**The arginine apex contacts the membrane interface when Syt1 C2AB is membrane bound.** Shown in Fig. 1 is a model for Syt1 where the two C2 domains are attached to the vesicle membrane through a long linker. In the present work, we utilize a soluble fragment of Syt1 containing the C2A and C2B domains (residues 136–421). Site-directed spin labeling may be used to both monitor membrane contact and measure membrane depth, and our approach involves the incorporation of the spin-labeled side chain R1 into selected sites in the protein. As shown in Fig. 1b, membrane insertion of a region of the domain, in this case the first $Ca^{2+}$-binding loop of the C2A domain, produces a dramatic broadening in the EPR lineshape due to a reduction in the rotamers sampled by the R1 side chain.

To test for interactions in the arginine apex, spin labels were incorporated into three sites in the C2B domain that are depicted in Fig. 2a, and the EPR spectra were recorded in the absence of membrane and in the presence of PC:PS vesicles with and without $Ca^{2+}$. When the C2B domain is membrane associated in the presence of $Ca^{2+}$, changes in the lineshapes from spin labels at sites 285, 349, and 350 are observed. The spectra show a slight broadening and a decrease in the residual hyperfine interaction, Azz', that may be due to the approach of the labels to the membrane interface. To confirm that these labels approach the membrane, progressive power saturation was used to provide an estimate of the distance of the labels to the membrane interface. As shown in Fig. 2c, labels in the arginine apex lie close to the membrane interface in the presence of $Ca^{2+}$ and are about 2 Angstroms from a plane defined by the lipid phosphates on the aqueous side of the membrane (Supplementary Table 1). These three labels also tend to be near the interface in the absence of $Ca^{2+}$. This association may be facilitated by a weak association of the lysines forming the polybasic face of C2B (Fig. 2a) to the PC:PS membrane interface. In the absence of $Ca^{2+}$ and under the conditions of this measurement, there is likely a significant concentration of aqueous C2B domain that is in equilibrium with a population of membrane associated C2B.

When PIP$_2$ replaces PS in the membrane at equivalent charge densities, the EPR lineshapes and power saturation data also indicate that this region contacts the bilayer. As seen in Supplementary Fig. 1, site 350 makes a closer approach and site 285 is further away from the interface in PIP$_2$ than in PS. Interestingly, site 349 does not show strong evidence for contact, which likely reflects the altered orientation for the C2B domain when bound to a PIP$_2$ bilayer[8]. The labeled position that shows the clearest evidence for membrane contact is site 350, and unlike the case for the interaction with PC:PS the interaction to PIP$_2$ does not appear to be $Ca^{2+}$ dependent. This likely reflects the enhanced $Ca^{2+}$-independent membrane affinity of C2B to the PIP$_2$ bilayer relative to the bilayer containing PS[8]. Thus, we conclude that when C2AB is attached to the membrane interface, the arginine apex is also in contact with the bilayer surface.

**Membrane contact at the apex is driven by conserved arginine residues.** We then examined the effect of two sets of mutants,

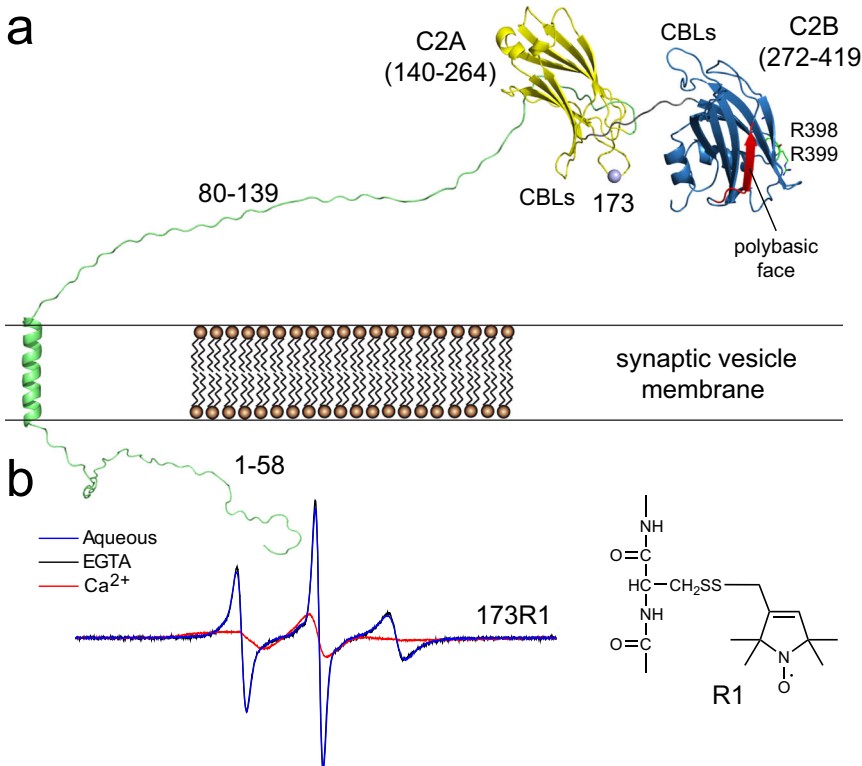

**Fig. 1 EPR spectroscopy is sensitive to membrane proximity and insertion of labeled Syt1. a** Model for Syt1 on the vesicle membrane surface showing position 173 in the 1st $Ca^{2+}$-binding loop (CBL) of the C2A domain (yellow). The position of the polybasic face (red) and R398, 399 side chains are shown for C2B (blue). **b** EPR spectra from an R1 label attached to site 173. In the presence of $Ca^{2+}$, the domain inserts into the bilayer, which alters the sampling of rotamers by the R1 side chain and broadens the EPR spectrum.

R398Q (RQ) and R398Q/R399Q (RQRQ) on the membrane depth parameters from the two most closely associating positions, 285 and 350, on the C2B domain of Syt1. Figure 3a shows the results of power saturation measurements for the wild-type C2AB versus the single RQ and double RQRQ mutations (Supplementary Table 2). Each arginine mutation shifts the apex further towards the aqueous phase or eliminates the membrane interaction, with the double RQRQ mutation showing the largest effects. For example, in the presence of PC:PS the RQRQ mutation shifts the position of 285R1 ~5 Å towards the aqueous phase. In PC:$PIP_2$, the RQRQ mutation eliminates the membrane interaction of 285R1. Shown in Fig. 3b are comparisons of the EPR spectra from site 285 with and without membrane for the wild-type C2AB and RQRQ mutation. The spectra obtained from site 285 in the presence of the RQRQ mutation are identical with and without membrane indicating that there is no membrane contact at this site in the presence of the RQRQ mutation. Similar trends in the depth data are seen for the RQ and RQRQ mutants in the absence of $Ca^{2+}$ (Supplementary Fig. 2).

**The arginine apex contacts membrane reconstituted SNARE proteins**. The arginine apex has been observed to interact with the soluble SNARE complex by X-ray crystallography[10]. We also examined the EPR spectra from the three spin-labeled sites at the apex of C2B to test for this interaction with SNARE complexes embedded in lipid bilayers. Shown in Fig. 4a are EPR spectra from sites 285, 349, and 350 in the presence of PC:PS membranes with or without membrane reconstituted SNAREs composed of full-length Syx, SNAP-25, and soluble Syb. The spectra for sites 285 and 350 have reduced normalized amplitudes and broadened hyperfine extrema (Azz') indicating that the motion and rotamer

sampling of the nitroxide at these sites is being sterically restrained by interactions with the SNAREs. We examined the rotamers available at these sites as predicted from the crystal structure (PBD ID: 5CCH) using the program MMM[19] and the result is shown in Fig. 4b. According to this model, the R1 label at all three sites should contact the SNAREs, with sites 350 and 285 having the greatest reduction in the available rotamers and site 349, which projects away from the SNAREs, having the least. This is consistent with the data in Fig. 4a and indicates that these EPR data in the presence of membrane reconstituted SNAREs are consistent with the model from crystallography.

**ATP or $PIP_2$ eliminate SNARE but not membrane interactions of the arginine apex**. Next, we tested the effect of the RQ mutations and mutations in SNAP-25 on the association of Syt1 with the SNAREs. Interactions between Syt1 C2AB and the SNAREs are likely driven by charge interactions, and we mutated three negatively charged residues D51, E52, and E55 to alanine in the N-terminal segment of SNAP-25. As seen in Fig. 4c, a comparison of the EPR spectra from spin-labeled sites in the Syt1 C2AB arginine apex show no differences between PC:PS membranes alone and PC:PS membranes containing SNAREs with this AAA mutant. The result indicates that the AAA mutation eliminates the Syt1 C2AB-SNARE interaction. Importantly, as shown below, this mutation somewhat reduces docking of secretory vesicles and their overall fusion, but it does not alter the $Ca^{2+}$ dependent release mode (fusion pore opening) and $Ca^{2+}$ stimulated fusion increase. Figure 4d compares normalized amplitudes of spectra obtained from 285R1 and 350R1 in the presence of membrane reconstituted SNAREs, membrane reconstituted SNAREs having the AAA mutation, as well as membrane

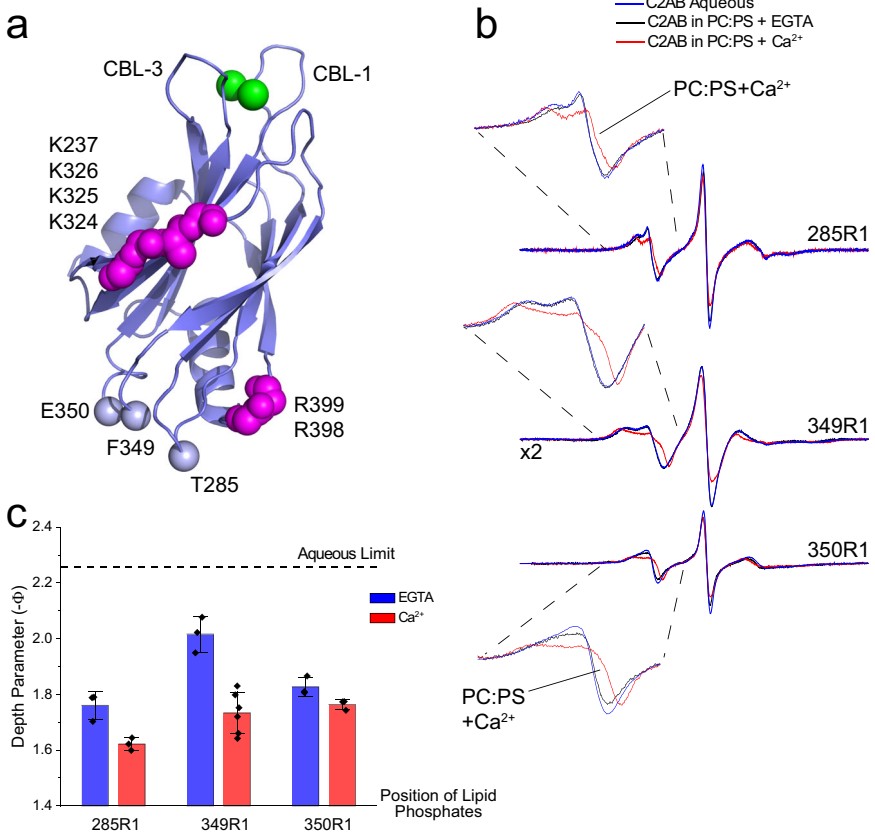

**Fig. 2 The arginine apex of C2B contacts the membrane interface. a** Model of C2B showing conserved arginine residues in the apex, lysine residues in the polybasic face and the three sites to which R1 was attached. The arginine apex is on the opposite surface of the domain from two $Ca^{2+}$-binding loops (CBL) that insert into bilayers in the presence of $Ca^{2+}$. **b** EPR spectra from the three labeled sites in the absence of membranes or the presence of POPC:POPS (80:20) lipid vesicles. **c** Membrane depth parameters obtained with and without $Ca^{2+}$ for labels near the apex. The bars indicate the mean value with ±standard deviation. Three independent samples and power saturation measurements were made, except for 349R1 with $Ca^{2+}$ where six samples were measured.

reconstituted SNAREs in the presence of the RQ and RQRQ mutations. Both single and double RQ mutations, as well as the AAA mutation, yield normalized amplitudes like those obtained for membranes lacking the SNAREs (composed only of PC:PS), indicating that the Syt1/SNARE interaction no longer takes place. The addition of 1 mM ATP/$Mg^{2+}$ (a typical cytoplasmic concentration) also eliminates interactions between C2AB and the membrane reconstituted SNAREs as does the presence of 5 mol% $PIP_2$ in the membrane (Fig. 4d).

As seen in Figs. 3 and 4, both the membrane and SNARE interactions of the arginine apex are sensitive to the RQ and RQRQ mutations. Thus, both interactions require these conserved arginine residues and are likely driven by electrostatics. However, the SNARE interaction is eliminated by $PIP_2$, whereas the membrane interaction is not. SNARE interactions are also eliminated by ATP, which is also not the case for the membrane interaction (Supplementary Fig. 3).

**Mutations in the arginine apex do not alter fusion probability but alter the rate of fusion pore opening.** The data in Figs. 3 and 4 indicate that conserved arginine residues in the apex of the C2B domain facilitate membrane contact of the apex under the conditions where fusion takes place. To better define their role in the fusion process, we used a single-particle fusion assay to examine fusion between chromaffin secretory granules purified from PC12 cells (also referred to as dense core vesicles, DCVs) and reconstituted planar supported membranes containing recombinant

syntaxin-1a and dSNAP-25 (dodecylated SNAP-25). These purified chromaffin granules were previously shown to be composed of the same molecular machinery as synaptic vesicles purified from rat brain[20]. An important feature of this assay when investigating the machinery of secretion is the ease of engineering stable knockdowns into PC12 cells and thus producing cell-derived vesicles with syt1 depletions and no other changes.

The chromaffin granules are labeled by expressing the releasable content marker neuropeptide Y (NPY)-mRuby, and their binding to planar supported membranes in a SNARE dependent manner (schematic in Fig. 5a, left) can be observed using total internal reflection fluorescence microcopy (TIRFM)[21]. Granule binding is accompanied by an increase in fluorescence within the TIRF field, where a subset of the granules fuse after a variable time delay (Fig. 5b). Fusion is marked by a decrease in fluorescence followed by a rapid spike in fluorescence signal and then a further decay in signal (Fig. 5b and c). The initial decay in the signal is caused by the loss of NPY-mRuby from the granule as it diffuses into the cleft between the supported membrane and its supporting slide during an initial slow release phase. The rapid increase in fluorescence is a result of a rapid transfer of granule fluorescent content within the evanescent field of the TIRFM to the plane of the supported membrane and into the cleft under the supported membrane when the vesicle completely merges with the target membrane. This observable spike in intensity ($\Delta I_C$) is indicative of the amount of content that was left in the granule following the initial slow release phase. For fusion events where this initial slow phase lasts longer, less content will be available

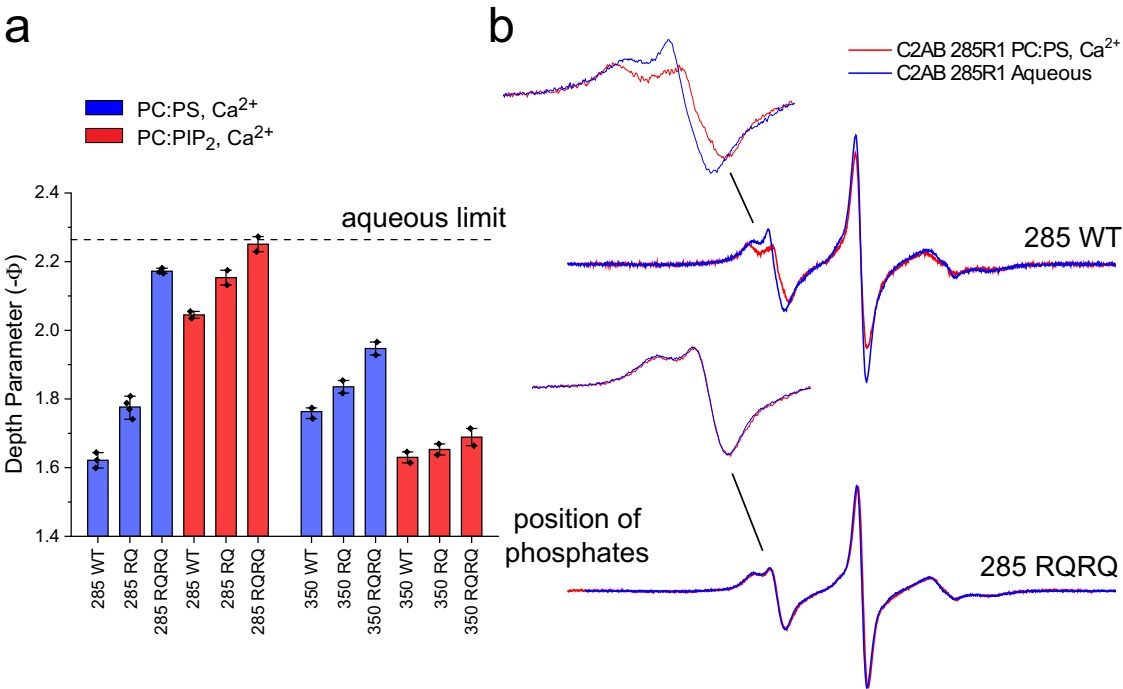

**Fig. 3 Mutating the arginine apex reduces or eliminates membrane contact by the arginine apex. a** power saturation indicates that the RQ and RQRQ mutations reduce or eliminate membrane contact. Vesicles were composed of POPC:POPS (80:20) or POPC:PIP$_2$ (95:5). The bars indicate mean values with outlying points. The points represent independent samples that were power saturated. Two independent measurements were made for POPC:PIP$_2$ samples. The 285R1 samples in POPC:POPS were repeated three times with the 285RQ sample repeated four times. The 350R1 WT sample in POPC:POPS was repeated three times. **b** EPR spectra obtained from site 285 for the pseudo wild-type (WT) and RQRQ mutant. EPR spectra are identical in the absence or presence of membranes for the RQRQ mutant. A depth parameter of 2 is obtained for an R1 label that lies ~4 Å from the lipid phosphate. When the label is further than 6 or 7 Å from the phosphates, the depth parameter becomes independent of label position.

during the rapid content transfer and the observed peak will be smaller. For events where the initial phase is short, more content will be available to be rapidly transferred within the evanescent field and the observed peak will be larger. As NPY-mRuby continues to diffuse away from the fusion site, the signal eventually decays. The origin of this signal has been previously described in detail[20,21] and is modulated by the lipid geometries that stabilize or destabilize fusion pores[22].

When granules are depleted of endogenous synaptotagmin isoforms, $Ca^{2+}$-dependent fusion is lost; however, addition of the soluble C2AB domain recovers the $Ca^{2+}$ dependence (Fig. 5d). The value of $\Delta I_C$ reflects the mode of contents release, i.e. the timing of the complete fusion event once pore opening has been initiated. As seen in Fig. 5c, this release mode is modulated by the presence of C2AB and 100 μM $Ca^{2+}$ when granules are depleted of endogenous synaptotagmins. The C2AB domain was previously shown to catalyze a structural transition in the SNARE complex by interactions mediated through the membrane bilayer[13]. This structural transition involves a change in SNARE orientation relative to the membrane, i.e., a transition from a trans-mimicking conformation to a cis-mimicking conformation, which can be monitored using site-directed fluorescence interference contrast microscopy (sdFLIC) (Fig. 5a, right)[13]. As seen in Fig. 5e this conformational change is dependent on C2AB and calcium. Combining the TIRF fusion and sdFLIC assays creates a powerful tool to examine secretory granule binding, fusion probability, content release, SNARE complex conformational changes, and kinetics of fusion, all under the same conditions. In the context of the various Syt1 mutations and their modulated interactions with membranes and SNAREs, this approach allows us to tease out the role of Syt1 in fusion.

Figure 5f shows the effects of three sets of mutants in C2B on granule binding, fusion efficiency, content release, structural changes in the SNAREs, and kinetics of fusion. Here, we examined a single arginine mutation in the apex (RQ), mutation of both arginines in the apex (RQRQ), or lysine mutations in the polybasic face (KAKA). As shown previously, in the absence of $Ca^{2+}$ granule binding is mediated by SNAREs and in the presence of $Ca^{2+}$ it is mediated by both SNAREs and the vesicle-resident calcium binding protein CAPS[21]. Consistent with this result, Fig. 5f shows that these mutations in C2B have no effect on granule binding. When the percentage of granule fusion events is examined, only mutations in the polybasic face (KAKA) depress the fusion efficiency, while mutations in the arginine apex have no effect. However, when we examine the timing of the fusion event (the release mode), the KAKA mutant has no effect while mutations in the arginine apex strongly affect the release mode. Thus, mutations in the apex strongly affect the transition from the slow to fast phase of content release, i.e., the transition from the opening to the dilation of the fusion pore. In contrast, as shown previously[13], mutations in the arginine apex produce only small perturbations in the conformation change of SNARE complex (measured by sdFLIC) compared to wild-type C2AB, while stronger effects on this orientational change are observed for the KAKA mutation. The polybasic face also had a greater effect on the kinetics of the delay time from granule binding to fusion, while the arginine apex mutations had only a slight effect.

As seen in Fig. 5f, the SNARE structural changes strongly correlate with the fusion efficiency and kinetics, and it is interesting to note that the effects of these C2B mutants track with their effect on C2AB membrane affinity. The KAKA mutant, which had the largest effect on SNARE structural changes and

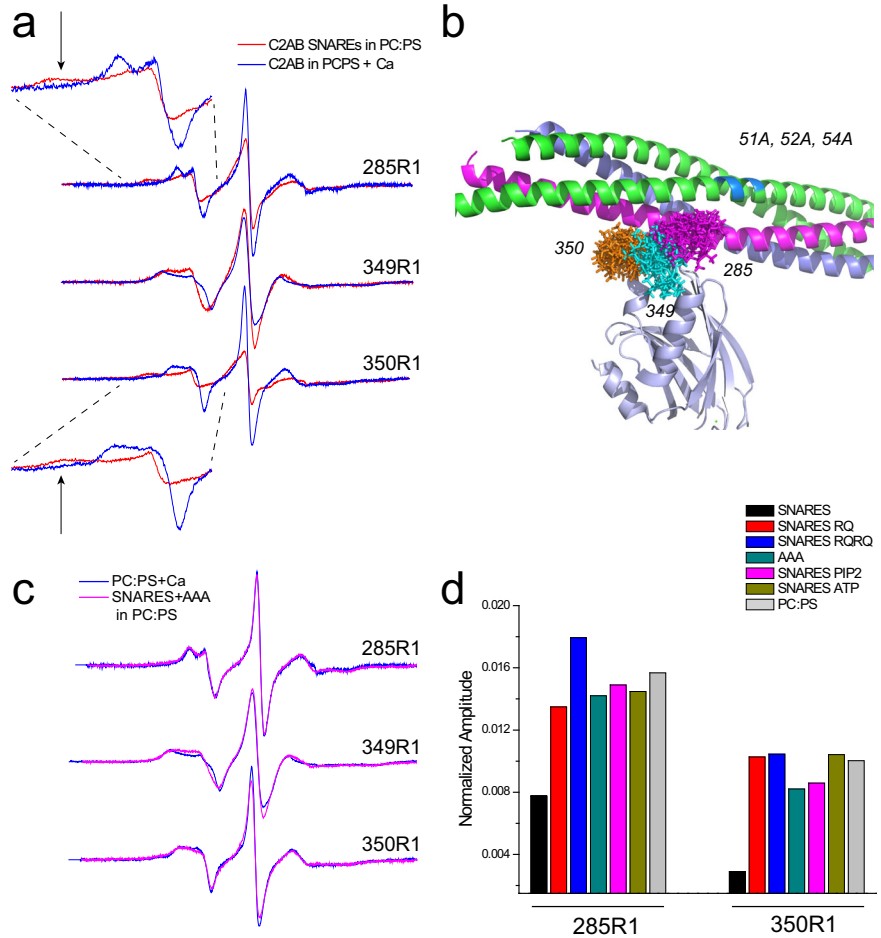

**Fig. 4 The arginine apex contacts membrane reconstituted SNAREs, but ATP or PIP$_2$ in the bilayer, eliminate the interaction. a** EPR spectra in the absence (blue trace) or presence (red trace) of membrane reconstituted SNARE complex composed of full-length Syx, SNAP-25, and soluble Syb. The broadened components in the spectra (arrows) are due to incompletely averaged axial components of the hyperfine tensor and indicate tertiary contact of the R1 side chain with SNAREs. **b** Model (PDB ID 5CCH) with available rotamers for the R1 side chains at the three sites. **c** Comparison of EPR spectra with and without membrane reconstituted SNAREs containing the AAA mutation in SNAP-25. **d** Normalized amplitudes for EPR spectra from sites 285R1 and 350R1. The normalized amplitudes provide a semi-quantitative measure of R1 motion where larger amplitudes indicate loss of R1 tertiary contact with the SNAREs.

fusion, is known to reduce the membrane affinity of C2AB in the presence of PIP$_2$[8]. In contrast, the membrane affinity of C2AB was measured for the RQRQ mutation in the apex and was found to produce no significant change in affinity (Supplementary Fig. 4). The data in Fig. 5f indicate that the membrane binding of Syt1 C2AB is necessary to initiate fusion, but that the membrane interactions made by the arginine apex are required to for the proper timing of the fusion event and the rapid opening of the fusion pore.

**The C2B domain makes simultaneous membrane contact at the arginine apex, the polybasic face and the Ca$^{2+}$-binding loops.** The data shown in Fig. 2 indicate that the arginine apex contacts negatively charged membranes whenever the C2B domain is membrane associated. However, other regions of C2B also make membrane contact. In the presence of PIP$_2$ and under the same conditions that attach the arginine apex, the Ca$^{2+}$-binding loops penetrate and the polybasic face interacts with PIP$_2$[8]. These interactions are evident in the EPR spectra shown in Fig. 6a, where the EPR spectra and power saturation data show that these regions of C2B are interacting with the membrane interface and with PIP$_2$ (Fig. 6b and Supplementary Table 3).

Inspection of the spectra from sites 304 and 329 indicates that for virtually the entire population of C2B, these regions are membrane associated or interacting with PIP$_2$. Multiple components in the spectra that would indicate an equilibrium between aqueous and membrane associated C2B are not seen. Such an equilibrium would be more difficult to distinguish in the spectra from the apex (Fig. 2), where spectra corresponding to bound and unbound apex are similar. Nonetheless, the presence of a mixture of membrane associated and aqueous states is not obvious. Thus, a large fraction of C2B must undergo simultaneous membrane contact at these sites. Shown in Fig. 6c are the allowable rotamers of the nitroxide spin label at each of the membrane interacting sites. It is not possible to simultaneously satisfy these interactions with a flat planar bilayer interface or two parallel bilayer surfaces; however, these interactions can be satisfied if the C2B domain interacts with a curved membrane interface.

**Removal of PIP$_2$ alters granule binding, fusion, fusion pore opening, and SNARE conformational changes.** The association of the polybasic face with the membrane interface seen in Fig. 6a is driven by PIP$_2$, and we tested the effect of PIP$_2$ on granule fusion and SNARE conformation as described above for the C2B

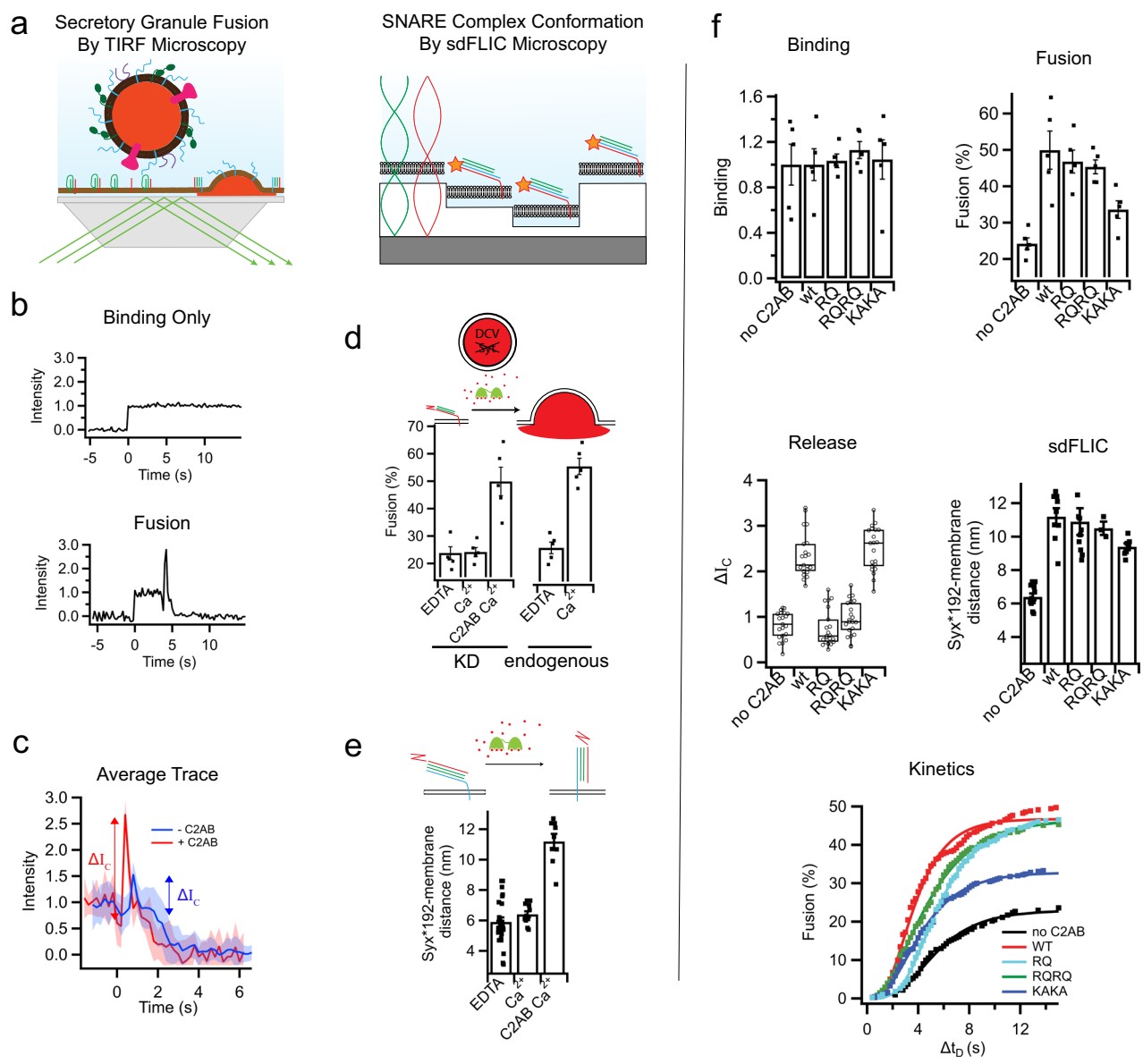

**Fig. 5 The effects of synaptotagmin mutations in C2B on secretory granule fusion and SNARE orientation. a** Single granule supported membrane TIRF fusion assay (left) and sdFLIC microscopy of the ternary SNARE complex (right). **b** Fluorescence intensity traces of single granules interacting with supported membranes in a TIRF microscope. After a sudden increase in fluorescence, indicating binding, the intensity stays constant if the granule never fuses (top) or the trace shows a characteristic dip and peak after different delay times (bottom) if the granule membrane fuses with the reconstituted supported membrane. **c** Averaged traces (20 events per trace) showing a fast mode of fusion in the presence of C2AB (red) or slow mode of fusion in the absence of C2AB (blue). Both conditions were in the presence of 100 µM Ca$^{2+}$. **d** Granules depleted of synaptotagmins (Syt1 and Syt9) do not fuse in response to calcium. 0.4 µM soluble C2AB stimulates fusion of synaptotagmin knockdown granules to a similar level as granules containing endogenous synaptotagmin in the presence of 100 µM Ca$^{2+}$ (membrane composition was 32:32:15:20:1 bPC:bPE:bPS: Chol:PIP$_2$). **e** Structural changes in the orientation of the ternary SNARE complex in the presence of calcium and 0.4 µM C2AB. **f** The effects of RQ, RQRQ, or KAKA mutations in C2AB on granule binding, fusion, release characteristics, SNARE orientation, and fusion kinetics in the presence of 100 µM Ca$^{2+}$ (membrane composition was 32:32:15:20:1 bPC:bPE: bPS:Chol:PIP$_2$). The bar plots show mean ± standard error or the mean. The boxplots show mean, 2nd and 3rd quartile, and outliers of the data.

mutants. As shown in Fig. 6d, removal of PIP$_2$ in the presence of 100 µM Ca$^{2+}$ reduces granule binding, fusion efficiency, slows the fusion timing or release mode, alters SNARE conformation, and reduces fusion kinetics. The loss of PIP$_2$ will reduce C2AB affinity and should have a similar effect as the KAKA mutant. But the loss of PIP$_2$ also produces an effect on the opening of the fusion pore that is similar to that of the RQ and RQRQ mutants (Fig. 5f). The change in calcium stimulated granule binding mediated by PIP$_2$ was previously demonstrated to be dependent on CAPS, which is a membrane associated calcium binding protein with a PIP$_2$ specific PH domain[21].

**The SNAP-25 AAA mutation has no effect on granule contents release and the opening of the fusion pore.** As shown above in Fig. 4c, d, mutation of SNAP-25 to replace three negatively charged residues (the SNAP-25 AAA mutation) eliminated the SNARE binding of the arginine apex. We tested to see what effect these mutations would have on the granule docking and fusion and the opening of the fusion pore. As seen in Fig. 6e, in the presence of Ca$^{2+}$ the AAA mutation reduces, but does not eliminate, granule binding and fusion efficiency. However, the effects of the AAA mutation are also seen in the absence of Ca$^{2+}$ or C2AB, indicating that they are not mediated by C2AB. In

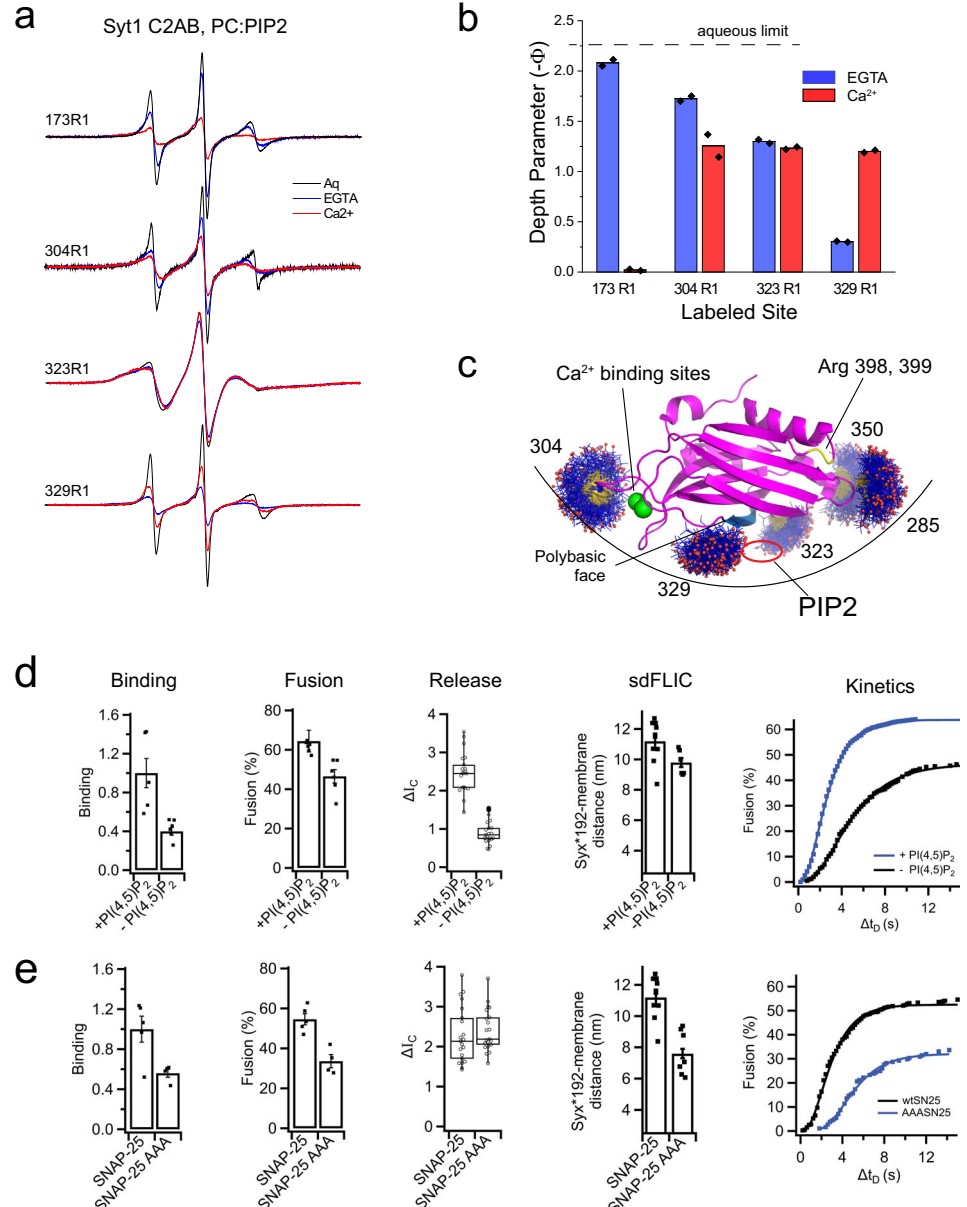

**Fig. 6 The C2B domains makes simultaneous membrane contact at multiple sites that cannot be explained by docking to a planar bilayer surface. a** EPR spectra in an aqueous state compared with spectra in the presence of POPC:PIP$_2$ (95:5) bilayers in the presence of EGTA or Ca$^{2+}$. **b** Membrane depth parameters indicate that site 173 inserts into the PIP$_2$ bilayer in the presence of Ca$^{2+}$. Sites on C2B including 304, 323, and 329 also contact with bilayer, with or without Ca$^{2+}$. Bar plots show mean values from the power saturation of two samples. **c** Allowable rotamers for labeled sites that contact the membrane interface including two near the arginine apex. The membrane contact seen at these sites could be accommodated by a curved membrane surface. This would place a PIP$_2$ headgroup in a binding pocket at the polybasic face. **d** The effect of PIP$_2$ on granule binding, fusion, release profile, SNARE orientation, and fusion kinetics in the presence of 100 μM Ca$^{2+}$ (membrane composition was 25:25:15:30 bPC:bPE:bPS:Chol and either 5% PI or 4% PI with 1% PIP$_2$). **e** The effect of the AAA mutation in SNAP-25 on granule binding, fusion, release profile, SNARE orientation, and fusion kinetics in the presence of 100 μM Ca$^{2+}$ (membrane composition 32:32:15:20:1 bPC:bPE:bPS:Chol:PIP$_2$). The bar plots show mean ± standard error of the mean. The boxplots show mean, 2nd and 3rd quartile, and outliers of the data.

contrast to the RQ and RQRQ mutations (Fig. 5f), the AAA mutation has no effect on the release mode or timing of the fusion event. Since this mutation eliminates the SNARE interactions made by the C2B arginine apex, SNARE interactions of the apex are not necessary to mediate changes in Ca$^{2+}$-dependent fusion pore opening.

## Discussion

Despite its central importance for synchronous neurotransmitter release, the molecular mechanisms underlying Syt1 function are

presently unclear. Molecular models for Syt1 in association with SNAREs have been generated by X-ray crystallography, and they suggest that the Ca$^{2+}$ regulatory event involves a control of the SNAREs by a direct interaction with Syt1[10,11]. However, other data indicate that the interactions of Syt1 with SNAREs are weak, heterogeneous, and are of secondary importance to the interactions made by Syt1 to membranes[12].

In the present work, we examined a region on the C2B domain that lies opposite the Ca$^{2+}$-binding loops that contains a pair of conserved arginine residues at sites 398 and 399. Mutating this

pair of residues to glutamine (RQRQ mutation) is known to have a significant effect on synchronous neurotransmitter release and excitatory postsynaptic potentials[17], and our initial goal was to determine whether the effect of these mutations was acting at the level of a membrane or direct SNARE interaction. As shown in Fig. 2 and Supplementary Fig. 1, the arginine apex associates with membranes whenever Syt1 C2AB is membrane associated. The apex also associates with membrane reconstituted SNAREs in a manner that is consistent with molecular structures generated by crystallography, and both membrane and SNARE interactions are weakened or eliminated by the RQ and RQRQ mutations. However, the data presented here indicate that only the membrane interactions of the C2B apex occur under conditions where fusion takes place.

The interaction made by the arginine apex with membrane reconstituted SNAREs is eliminated when $PIP_2$ is present in the membrane (Fig. 4d), or when ionic conditions mimic those expected within the cell. In addition, mutation of three basic residues within SNAP-25 (the AAA mutation) also eliminates the Syt1-SNARE interaction (Fig. 4c, d). However, the AAA mutation does not abolish membrane docking or fusion (Fig. 6e), and it has no effect on the rate of fusion pore expansion during the fusion event, indicating that interactions between the C2B arginine apex and SNAREs are not involved in $Ca^{2+}$-triggered neuropeptide release. In contrast, when $PIP_2$ is present in the membrane, the apex of C2B contacts membranes under all conditions we examined and is progressively weakened by the RQ and RQRQ mutations. These mutations in the apex have a profound effect on the rate at which the fusion pore opens and the concentration of vesicle content that is released quickly into the TIRF field (Fig. 5f). These data demonstrate that the membrane interactions of the apex of Syt1, and not the SNARE interactions, regulate the fusion pore expansion.

Previous work has shown that Syt1 C2AB binds preferentially to membranes rather than SNAREs when $PIP_2$ is present, an interaction that is mediated by the polybasic face of C2B[8,12]. Here, the KAKA mutation in the polybasic face of C2B did not alter the opening of the fusion pore (Fig. 5f), rather it altered SNARE conformational changes and fusion probability and kinetics. The result indicates that the apex plays a role in Syt1 function that is distinct from that of the polybasic face and that there are at least two molecular roles played by C2B that are important for fusion.

Isoforms of synaptotagmin are known to have different behaviors with respect to $Ca^{2+}$-sensitivity and synchronous release[23]. The arginine apex appears to be important in controlling the release mode of synaptotagmins, as shown by a comparison of the behavior of the Syt1 RQ and RQRQ mutants with that of synaptotagmin-7 (Syt7). Using the same single-particle fusion assay as that used here, Syt7 demonstrates a release mode that is significantly different than WT Syt1, where the timing of the fusion event appears to be slowed[24]; however, the release mode for Syt7 is similar to that seen here for either the RQ or RQRQ mutations of Syt1 (Fig. 5f). A comparison of the sequence of the arginine apex for Syt1 and Syt7 shows that Syt7 is RQ rather than RR, and thus is identical in this regard to the Syt1 RQ mutant. We speculate that Syt7 and the RQ mutant of Syt1 look similar in terms of the rate of fusion pore opening because both proteins have the same residues in this apex. It should be noted that the RQ and RQRQ mutations do not abolish release, they simply fail to accelerate the timing of the complete fusion event that would be seen with WT Syt1, $Ca^{2+}$, and $PIP_2$. As shown in Fig. 5f, these release events resemble those where Syt1 is absent and can therefore be considered to constitute cases where pore expansion proceeds uncontrolled or unregulated.

How might membrane interactions of the apex of C2B act to mediate membrane fusion and alter the characteristics of the fusion pore? In previous work, membrane interactions made both by the arginine apex and the $Ca^{2+}$-binding loops on opposite surfaces of the domain suggested a model where C2B bridged across two parallel bilayer planes, perhaps ultimately functioning to shorten the vesicle-plasma membrane distance in the presence of $Ca^{2+}$ [15,18]. However, the data presented here indicate that in the presence of $PIP_2$, the polybasic face also contacts the $PIP_2$ headgroup (Fig. 6a, b), so that three regions of the C2B domain make membrane contact. The EPR lineshapes indicate that the interaction of the polybasic face and the $Ca^{2+}$-binding loops are stable interactions that are not transient. This is more difficult to determine for the arginine apex, but nonetheless simultaneous membrane interactions of each of these three regions must occur for at least some period. It is not possible to dock the C2B domain to either one planar bilayer surface or to two parallel planar bilayer surfaces so that contact in all three regions is satisfied. But simultaneous interactions in all three regions of C2B can be satisfied if the domain is interacting with a curved bilayer interface (Fig. 6c). Indeed, recent simulations suggest that mixtures of lipids likely to mediate fusion can form punctate surfaces with regions of negative curvature[25], which might accommodate this type membrane interaction of C2B.

Shown in Fig. 7 is a model for the membrane interactions made by Syt1 that incorporates what is known about the molecular interactions made by C2B and the effect of mutations in the arginine apex on granule binding and fusion. In this model, Syt1 is interacting at the negatively curved surface that is present following the formation of a hemifusion intermediate. As the fusion pore forms, the C2B domain of Syt1 interacts with this negatively curved surface making contact at the $Ca^{2+}$-binding loops, arginine apex and polybasic face. In doing so, the C2B domain of Syt1 helps mediate the transition during a fusion event to a fully open pore. It might mediate this transition by either lowering an energy barrier for pore expansion or by preventing the system from getting trapped at an early intermediate state during the pore opening[26]. By weakening or eliminating membrane contact at the apex with either the RQ or RQRQ mutations, the C2B domain no longer assumes the correct orientation across the cytoplasmic side of the fusion pore, and the mode of contents release or progression to a fully open fusion pore is slowed (Fig. 5f). Other than the RQ and RQRQ mutations, $PIP_2$ is the other component that is seen to strongly influence the release mode or pore opening when it is removed (Fig. 6e). Interestingly, the polybasic face is aligned facing into a region of high negative curvature in this model. Since it interacts with $PIP_2$, this lipid would be driven into an interface of negative curvature. $PIP_2$ is a lipid that alone will tend to form micelles[27,28], and its position within the negative curve of the growing fusion pore would not be energetically favored. Conceivably, the presence of $PIP_2$ in this region of the pore, driven by strong electrostatic interactions with the polybasic face of C2B, might act to destabilize an initial state in the fusion pore and drive it to a fully open state.

In summary, the data presented here are consistent with a model for the role of Syt1 where the C2B domain interacts with the negatively curved membrane surface during the initial stages of fusion to mediate expansion of the fusion pore. Using both EPR spectroscopy and single-particle fusion experiments we demonstrate that the arginine apex of C2B helps control the opening or dilation of the fusion pore and mediate the transition between the initial pore opening and a fully open fusion pore. We demonstrate that the C2B domain interacts both with membranes and membrane associated SNAREs; however, only the membrane interaction takes place under the conditions where fusion occurs. As a result, the expansion of the fusion pore is mediated by a membrane interaction of the C2B arginine apex. Mutations that lie in the polybasic face of C2B alter the fusion probability but

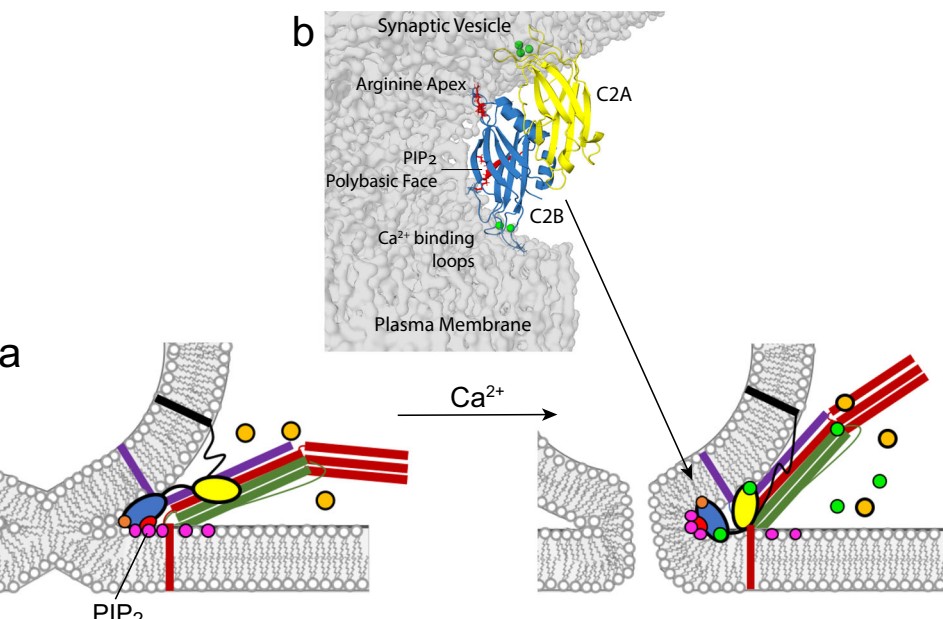

**Fig. 7 Model for the membrane interactions of Syt1.** In **a** after docking to the bilayer and in the presence of ATP/Mg$^{2+}$ (orange) the SNARE complex, syntaxin (red), SNAP-25 (green), synaptobrevin (purple), and Syt1 C2B (blue), promote partial lipid mixing by either zippering or contacting plasma membrane PIP$_2$ at the arginine apex (orange) and the polybasic face (red). Upon calcium influx (green), Syt1 C2A inserts into the synaptic vesicle membrane, PS, and C2B reorients, binding the Ca$^{2+}$-binding loops at a 3rd membrane contact point. The C2B domain of Syt1 might position at the site of fusion as shown in **b**. This orientation would sequester PIP$_2$ into the strained membrane stalk, further destabilizing the membrane and promoting pore opening. The C2A domain is shown inserted into the vesicle surface. This orientation of C2A is preferred in full-length Syt1[15]; however, an interaction of C2A with the plasma membrane may also take place. The Ca$^{2+}$-binding loops of C2B are shown directed towards the plasma membrane surface because C2B has a high affinity towards PIP$_2$ containing membranes in a low Ca$^{2+}$ state and the loops are likely to insert into the plasma membrane[8].

have no effect on the expansion of the fusion pore once fusion occurs, indicating that there are at least two molecular roles for Syt1 in triggering fusion.

## Methods
**Expression and purification of Syt1 and SNAREs proteins**. Expression and purification of C2AB (residues 136–421) from *Rattus norvegicus* was carried out using a pGEX-KG construct having an N-terminal GST-tag where the native cysteine at residue 277 was mutated to an alanine. For electron paramagnetic resonance (EPR) measurements the single site of interest was mutated to cysteine using Polymerase Incomplete Primer Extension (PIPE) site-directed mutagenesis to attach a spin label at sites: M173C, T285C, V304C, L323C, T329C, F349C, and E350C. Sites T285, F349, and E350 were chosen to be outward facing on loops near the arginine apex, where labels would not interfere with arginine side chains or the folding of the domain. The remaining sites were chosen based upon previous work to examine the membrane interactions of C2AB. To neutralize the arginine apex, R398Q (RQ) or R398QR399Q (RQRQ) mutations were also introduced to the cys-free construct, the T285C plasmid, and the E350C plasmid. A list of the mutagenesis primers used in this study is provided in Supplementary Table 7.

The C2AB plasmid was transformed into BL21(DE3) cells on ampicillin containing plates, and a single colony was used to inoculate 50 mL precultures of Luria Bertani (LB) media with ampicillin (50 mg/mL) and grown overnight at 37 °C. Precultures were used to inoculate 1 L main cultures of LB which were grown at 37 °C. When an OD$_{600}$ of 0.8–1.0 was reached, the cells were induced using 0.1 mM IPTG and grown overnight at 20 °C. Cells were harvested by centrifugation at 12,000 rpm, 12 min, 4 °C, then resuspended in phosphate-buffered saline (PBS) containing 2 mM EGTA and 2% triton X-100. Protease inhibitors: 50 uM leupeptin, 20 units/mL aprotinin, and 0.8 mM AEBSF, and 0.01% benzonase nuclease were added to the suspension and the cells were lysed using a French press. Cell debris were removed by ultracentrifugation at 18,000 RPM for 30 min at 4 °C and the supernatant was collected and loaded onto a GSTPrep FF 16/10 column (Cytiva, Marlborough, MA) using an AKTA prime FPLC (GE Healthcare, Chicago, IL). The column was washed using Bead buffer (PBS, 1% tritonx100, 2 mM EGTA), Prep buffer (50 mM tris base, 150 mM NaCl, pH 8.4), then equilibrated with Cleavage buffer (50 mM tris base, 150 mM NaCl, 4 mM CaCl$_2$, pH 8.4).

For GST-tag cleavage and spin labeling, approximately one column volume of Cleavage buffer containing 5 mg thrombin and 3 mg of methanethiosulfonate spin label ((1-oxy-2,2,5,5-tetramethylpyrrolinyl-3-methyl)methanethiosulfonate), (Cayman Chemical, Ann Arbor Michigan) was manually injected into the column.

The column was then covered in foil and placed on a rocker at room temperature overnight. The protein was then eluted from the GSTPrep column using the AKTA prime. Three GSTtrap FF and 1 HiTrap benzamidine FF(HS) columns (SigmaMillipore, Burlington, MA) were attached in sequence after the GSTPrep column to separate the spin-labeled protein, the thrombin, and the GST tags. Protein was eluted with Elution buffer (50 mM tris, 750 mM NaCl, 25 mM EGTA).

The protein sample was further purified using ion exchange by first concentrating the samples using an Amicon Ultra-15 10 K concentrator (MilliporeSigma, Burlington, MA), performing ion exchange into SPA buffer (50 mM MOPS, 1 mM CaCl$_2$, 150 mM NaCl, pH 7.2) using a HiPrep 26/10 desalting column (SigmaMillipore, Burlington, MA), followed by ion exchange using a HiTrap SP HP column (SigmaMillipore, Burlington, MA). The protein was eluted under a gradient of SPA and SPB buffer (50 mM MOPS, 1 mM CaCl$_2$, 800 mM NaCl, pH 7.2) to isolate nucleic acid free protein. Each eluted fraction was examined for nucleic acid contaminants by measuring the 260/280 nm ratio using a NanoDrop spectrophotometer (ThermoFisher, Grand Island, NY). Ratios for pure protein were typically 0.45 or less and these fractions were collected and concentrated to ~50–100 μM and exchanged into physiological buffer (20 mM HEPES, 150 mM KCL, pH 7.4) using an Amicon 10 K concentrator.

The SNARE complex used in these experiments was derived from full-length cysteine free syntaxin, full-length cysteine free SNAP25A, with all four cysteines replaced by serine, and soluble synaptobrevin (residues 1–96), where the SNARE proteins were based upon sequences from *Rattus norvegicus*[29]. For the AAA mutant in the SNARE complex, the negatively charged positions D51, E52, E55 of cysless-SNAP25A were mutated to alanine. DNA sequencing for all mutations was verified by GENEWIZ DNA sequencing (South Plainfield, NJ). The sequences were cloned into pET28a vectors to yield N-terminally His6-tagged proteins sandwiched by thrombin cleavage sites. The recombinant SNAREs were expressed in BL21 (DE3) cells (Agilent Technologies, Santa Clara, CA) and purified by gravity column chromatography on Ni-NTA agarose resin (Thermo Fisher Scientific, Waltham, MA) and eluted by 400 mM imidazole in a buffer of 500 mM NaCl, 20 mM HEPES, pH = 7.4. Eluted fractions were digested by thrombin overnight at 4 °C, and subsequently purified by ion-exchange chromatography (MonoQ for syntaxin and SNAP-25, and MonoS for synaptobrevin). Finally, purified fractions were buffer exchanged into assembly buffer (20 mM HEPES, pH 7, 150 mM NaCl, 1 mM EDTA).

**SNARE complex assembly**. For SNARE complex assembly, purified individual components of SNARE proteins were mixed in a molar ratio of 1:1:1 in assembly buffer with the presence of 0.1% dodecylphosphocholine (DPC). Usually, SNAP-25 and synaptobrevin were mixed first in buffers without DPC, after raising DPC

concentration to 0.1%, the appropriate amount of syntaxin was added for overnight incubation at 4 °C. The reaction mixture was then purified with ion-exchange chromatography (MonoQ 10/100, GE Healthcare, Chicago, IL) to remove small quantities of unreacted monomers.

**Continuous wave EPR**. Experiments were performed using a Bruker X-Band EMX spectrometer (Bruker BioSpin, Billerica, MA) equipped with an ER 4123D dielectric resonator. All EPR spectra were recorded at room temperature using 100-G magnetic field sweep, 1-G modulation, and 2.0 mW incident microwave power. The CW measurements were performed on 4 to 6-µL samples in glass capillary tubes (0.60 mm inner diameter × 0.84 mm outer diameter round capillary; Vitro-Com, Mountain Lakes, NJ). The phasing, normalization, and subtraction of EPR spectra were performed using in-lab software written by David Nyenhuis. To assess the membrane binding, ~50–100 µM of spin-labeled C2AB was added to either charged LUVs at a 1:200 protein:lipid ratio (~10–20 mM lipid concentration) of 1-palmitoyl-2-oleoyl-glycero-3-phosphocholine (POPC):1-palmitoyl-2-oleoyl-sn-glycero-3-phospho-L-serine (POPS), (80:20) or POPC/L-α-phosphatidylinositol-4,5-bisphosphate, brain (PIP$_2$), (95:5). Vesicles containing PIP$_2$ were freshly prepared for each experiment and most POPC:POPS vesicle preparations were used within a day. To assess membrane and SNARE binding, cys free or AAA SNAREs were reconstituted at a 1:400 protein:lipid ratio of the same lipid compositions by dialysis in the presence of Bio-Beads (Bio-Rad Laboratories, Hercules, CA) into metal-free buffer. C2AB was added to proteoliposomes containing SNAREs at a 1:1:200 C2AB:SNAREs:lipid ratio. All lipids were obtained from Avanti Polar Lipids (Alabaster, AL). The EPR spectra were recorded in the presence of 1 mM Ca$^{2+}$, after the addition 1 mM ATP/Mg$^{2+}$, and after the addition of 4 mM EGTA. Progressive Power saturation of the EPR spectra[8] was performed by placing samples into gas-permeable TPX-2 capillaries. Each sample was run in the presence of N$_2$ or air (O$_2$) or Ni(II)EDDA to determine the membrane depth parameter, Φ. The spin label depth was estimated using the following expression: Φ = A[tanh(B(x − C)) + D], where x is the distance of the spin label from the phospholipid phosphate plane in the bilayer, and A, B, C, and D are empirically determined constants[30].

To assess membrane and SNARE binding, cys free or AAA SNAREs were reconstituted at a 1:400 protein:lipid ratio of the same lipid compositions by dialysis in the presence of Bio-Beads (Bio-Rad Laboratories, Hercules, CA) into metal-free buffer. C2AB was added to proteoliposomes containing SNAREs at a 1:1:200 C2AB:SNAREs:lipid ratio. All lipids were obtained from Avanti Polar Lipids (Alabaster, AL). The EPR spectra were recorded in the presence of 1 mM Ca$^{2+}$, after the addition 1 mM ATP/Mg$^{2+}$, and after the addition of 4 mM EGTA. Progressive Power saturation of the EPR spectra was performed as described previously[8]. Briefly, samples were placed in gas-permeable TPX-2 capillaries. Then each sample was run in the presence of air (O$_2$) or Ni(II)EDDA to determine the membrane depth parameter, Φ. The spin label depth was estimated using the following expression: Φ = A[tanh(B(x − C)) + D], where x is the distance of the spin label from the phospholipid phosphate plane in the bilayer, and A, B, C, and D are empirically determined constants[30].

**Fluorescent labeling**. His-tagged syntaxin-1a was reacted with an at least twofold molar excess of Alexa-546 in thoroughly degassed DPC-buffers (20 mM HEPES, pH 7.4, 500 mM NaCl, 0.1% DPC). Labeled proteins were separated from free dye via extensive wash after rebinding to the Ni-NTA column. Subsequently, eluted Alexa-labeled proteins were subjected to thrombin cleavage and then purified by size-exclusion chromatography (Superdex 200 10/300, GE Healthcare, Chicago, IL)[31].

**Reconstitution of SNAREs into proteoliposomes**. All SNARE proteins, acceptor complex and SNARE complexes, were reconstituted using sodium cholate. All lipid stocks except PIP$_2$ were kept in chloroform at −20 °C. The PIP$_2$ stock was kept in a mixture of chloroform and methanol (4:1). The desired lipids (using the compositions indicated) were mixed and organic solvents were evaporated under a stream of N$_2$ gas followed by vacuum desiccation for at least 1 h. The dried lipid films were dissolved in 25 mM sodium cholate in buffer (20 mM HEPES, 150 mM KCl, pH 7.4) followed by addition of an appropriate volume of the desired SNARE protein (s) (syx/dSNAP-25 for fusion experiments and syx*192/SNAP-25/syb1-96 for sdFLIC experiments) in their respective detergents to reach a final lipid/protein ratio of 4000 (sdFLIC and SytKD-DCV fusion) or 3000 (wt DCV fusion) for each protein. After 1 h of equilibration at room temperature, the mixture was diluted with buffer to reach a sodium cholate concentration of 16 mM, close to the critical micellar concentration. The sample was then dialyzed overnight against 1 L of buffer with 1 buffer change after ~4 h.

**Preparation of planar supported bilayers containing SNARE complexes**. Planar supported bilayers with reconstituted plasma membrane SNAREs were prepared by the Langmuir–Blodgett/vesicle fusion technique[32–34]. FLIC chips or quartz slides were cleaned by dipping in 3:1 sulfuric acid: hydrogen peroxide for 15 min using a Teflon holder. Slides were then rinsed thoroughly in Milli-Q water. The first leaflet of the bilayer was prepared by Langumir-Blodgett transfer directly onto the quartz slide using a Nima 611 Langmuir–Blodgett trough (Nima, Conventry, UK) by applying the lipid mixture of 80:20:3 bPC:Chol:DMPE-PEG-triethoxysilane

(DPS, Shearwater Polymers, Huntsville, AL) for experiments with SytKD DCVs or 70:30:3 bPC:Chol:DPS for experiments with wt DCVs from a chloroform solution. After allowing the solvent to evaporate for 10 min, the monolayer was compressed at a rate of 10 cm$^2$/min to reach a surface pressure of 32 mN/m. After equilibration for 5–10 min, a clean quartz slide was rapidly (68 mm/min) dipped into the trough and slowly (5 mm/min) withdrawn, while a computer maintained a constant surface pressure and monitored the transfer of lipids with head groups down onto the hydrophilic substrate. Proteoliposomes were incubated with the Langmuir–Blodgett monolayer to form the outer leaflet of the planar supported bilayer. A concentration of 77 mM total lipid in 1.2 mL total volume was used. After incubation of the proteoliposomes for 2 h the excess proteoliposomes were removed by perfusion with 10 mL of buffer (120 mM potassium glutamate, 20 mM potassium acetate, 20 mM HEPES, pH 7.4 for DCV fusion experiments or 150 mM KCl, 20 mM HEPES, 100 µM CaCl$_2$, pH 7.4 for sdFLIC experiments).

**Plasmids and shRNA constructs**. Plasmids and shRNA constructs used for preparations of DCVs have been previously described[21]. For simultaneous shRNA knockdown of multiple Syt isoforms, a modified pLKO.5 vector containing shRNA expression cassettes targeting Syt1 (TRCN0000093258) and Syt9 (TRCN0000379591) from Mission shRNA plasmids (MilliporeSigma, Burlington, MA) was used.

**Cell culture**. Wild-type pheochromocytoma cells (PC12) and a PC12 cell line stably expressing a Syt1-Syt9 double knockdown shRNA cassette were cultured as previously described[21] on 10 cm plastic cell culture plates at 37 °C in 10% CO$_2$ in Dulbecco's Modified Eagle Medium (DMEM) High Glucose 1 × Gibco supplemented with 10% horse serum (Cellgro), 10% calf serum (Fe$^+$) (Hyclone), and 1% penicillin/streptomycin mix. Medium was changed every 2–3 days and cells were passaged after reaching 90% confluency by incubating 5 min in HBSS and re-plating in fresh medium. Cells were transfected with NPY-Ruby by electroporation using an Electro Square Porator ECM 830 (BTX, Holliston, MA). After harvesting and sedimentation, cells were suspended in a small volume of sterile cytomix electroporation buffer[58] (120 mM KCl, 10 mM KH$_2$PO$_4$, 0.15 mM CaCl$_2$, 2 mM EGTA, 25 mM HEPES-KOH, 5 mM MgCl$_2$, 2 mM ATP, 5 mM glutathione, pH 7.6) and then counted and diluted to ~14 × 10$^6$ cells/mL. 700 µL of cell suspension (~10 × 10$^6$ cells) and 30 µg of DNA were placed in an electroporator cuvette with 4 mm gap and two 255 V, 8 ms electroporation pulses were applied. Cells were then transferred to a 10-cm cell culture dish with 10 mL of normal growth medium. NPY-Ruby transfected cells were cultured under normal conditions for 3 days after transfection and then used for fractionation.

**DCV purification**. DCVs were purified using an iso-osmotic density gradient[21]. PC12 cells (15-30 10-cm plates depending on experiments) were scraped into PBS, pelleted by centrifugation, resuspended, and washed once in homogenization medium (0.26 M sucrose, 5 mM MOPS, and 0.2 mM EDTA). Following resuspension in (3 ml) homogenization medium containing protease inhibitor (Roche Diagnostics), the cells were cracked open using a ball bearing homogenizer with a 0.2507-inch bore and 0.2496-inch diameter ball. The homogenate was then spun at 4000 rpm (1000 × g), 10 min at 4 °C in fixed-angle microcentrifuge to pellet nuclei and larger debris. The postnuclear supernatant (PNS) was collected and spun at 11,000 rpm (8000 × g), 15 min at 4 °C to pellet mitochondria. The post-mitochondrial supernatant (PMS) was then collected, adjusted to 5 mM EDTA, and incubated 10 min on ice. A working solution of 50% Optiprep (iodixanol) (5 vol 60% Optiprep: 1 vol 0.26 M sucrose, 30 mM MOPS, 1 mM EDTA) and homogenization medium were mixed to prepare solutions for discontinuous gradients in Beckman SW55 tubes: 0.5 mL of 30% iodixanol on the bottom and 3.8 mL of 14.5% iodixanol, above which 1.2 ml EDTA-adjusted PMS was layered. Samples were spun at 45,000 rpm (190,000 × g) for 5 h. A clear white band at the interface between the 30% iodixanol and the 14.5% iodixanol was collected as the DCV sample. The DCV sample was then extensively dialyzed (2–3 buffer changes) in a cassette with 10,000 kD molecular weight cutoff (24–48 h, 3 × 5 L) into the fusion assay buffer (120 mM potassium glutamate, 20 mM potassium acetate, 20 mM HEPES, pH 7.4).

**Site-directed fluorescence interference contrast (sdFLIC) microscopy**. The principle of site-directed fluorescence interference contrast (FLIC) microscopy and the experimental setup as used in this work, has been described[31]. A membrane containing protein with specifically labeled cysteines is supported on a patterned silicon chip with microscopic steps of silicon dioxide. The fluorescence intensity depends on the position of the dye with respect to the standing modes of the exciting and emitting light in front of the reflecting silicon surface. The position is determined by the variable-height 16 oxide steps and the constant average distance between dye and silicon oxide[35].

Images were acquired on a Zeiss Axiovert 200 or Axio Observer 7 fluorescence microscope (Carl Zeiss, Thornwood, NY) with a mercury lamp as a light source and a ×40 water immersion objective (Zeiss; N.A. = 0.7). Fluorescence was observed through a 610-nm band-pass filter (D610/60; Chroma, Battleboro, VT) by a CCD camera (DV-887ESC-BV; Andor-Technologies, Belfast UK). Exposure

times for imaging were set between 40 and 80 ms, and the excitation light was filtered by a neutral density filter (ND 1.0, Chroma) to avoid photobleaching.

During sdFLIC experiments, we acquired 4–6 images, 20–30 min after buffer changes, for each membrane condition of one supported membrane. From each image, we extracted 100 sets of 16 fluorescence intensities and fitted the optical theory with the fluorophore-membrane distance as fit parameter. The standard deviation of these ~400–600 results were usually in the order of 1 nm. The optical model consists of five layers of different thickness and refractive indices (bulk silicon, variable silicon oxide, 4 nm water, 4 nm membrane, bulk water), which we kept constant for all conditions[31,36,37]. The reported errors for the absolute membrane distance are the standard errors from at least three repeats. Not included in these errors are systematic errors due to different membrane thicknesses or membrane-substrate distances between different lipid conditions and a systematic underestimation of the residue-membrane distance from 10–20% of protein that is trapped on the substrate proximal side of the supported bilayer. The reported errors after buffer changes or the addition of C2AB are the standard errors of the detected distance changes from at least three repeats for each condition. Based on previous experiments with polymer supported bilayers we estimate the systematic uncertainty for the measured absolute distance to be ~1–2 nm[36]. Statistics for sdFLIC data may be found in Supplementary Table 4.

**Total internal reflection fluorescence (TIRF) microscopy**. Experiments examining single-vesicle docking and fusion events were performed on a Zeiss Axiovert 35 fluorescence microscope (Carl Zeiss, Thornwood, NY), equipped with a ×63 water immersion objective (Zeiss; N.A. = 0.95) and a prism-based TIRF illumination. The light source was an OBIS 532 LS laser from Coherent Inc. (Santa Clara CA). Fluorescence was observed through a 610 nm band-pass filter (D610/60; Chroma, Battleboro, VT) by an electron multiplying CCD (DU-860E; Andor Technologies, Belfast UK). The prism-quartz interface was lubricated with glycerol to allow easy translocation of the sample cell on the microscope stage. The beam was totally internally reflected at an angle of 72° from the surface normal, resulting in an evanescent wave that decays exponentially with a characteristic penetration depth of ~100 nm. An elliptical area of 250 × 65 µm was illuminated. The laser intensity, shutter, and camera were controlled by a homemade program written in LabVIEW (National Instruments, Austin, TX).

**Single DCV fusion assay**. Acceptor t-SNARE protein containing planar supported bilayers were washed with fusion buffer containing EDTA or divalent metal $Ca^{2+}$ as indicated in text. They were then perfused with DCV (50–100 µL depending on preparation) diluted into 2 mL of fusion buffer (120 mM potassium glutamate, 20 mM potassium acetate, 20 mM HEPES, pH 7.4) with additions of 100 µM EDTA, or 100 µM $Ca^{2+}$ with or without 0.4 µM C2AB as indicated in text. The fluorescence from DCVs was recorded by exciting with the 532 nm laser and using a EMCCD camera. After injection of the DCV sample, the microscope was focused within no more than 30 s and then a total of 5000 images were taken with 200 ms exposure times and spooled directly to the hard drive.

Single-vesicle fusion data were analyzed using a homemade program written in LabVIEW (National Instruments, Austin, TX). Stacks of images were filtered by a moving average filter. The maximum intensity for each pixel over the whole stack was projected on a single image. Vesicles were located in this image by a single-particle detection algorithm described in[38]. The peak (central pixel) and mean fluorescence intensities of a $5 \times 5$ pixel$^2$ area around each identified center of mass were plotted as a function of time for all particles in the image series. Docking was quantified by determining the number of vesicles that bound to the surface during the 10-min experiment after their addition to the supported membrane and normalizing relative to a standard condition.

The fusion efficiency was determined from the number of vesicles that underwent fusion within 15 s after they docked relative to the total number of vesicles that docked. The fusion kinetics was determined by measuring the time delay between time of docking and onset of fusion for each fusing vesicle. The resulting cumulative distribution function was normalized by the fusion efficiency. Results are reported as mean ± standard errors from five repeats of the experiments. Release modes of individual fusion events were quantified by normalizing the peak intensity trace originating from vesicles by their intensity during the docked state. The intensity increase $\Delta I_C$ was determined from single vesicles and graphed as boxplots (center line: median, box defines upper and lower quartiles, whiskers illustrate min and max) with raw data. Statistics for single-vesicle fusion data may be found in Supplementary Tables 5 and 6.

**Software**. Fluorescence microscopy setups were controlled by custom built software written in LabView 2016 for Windows (National Instruments, Austin, TX). Single-vesicle fusion data were analyzed using custom built software written in LabView 2016 for Windows (National Instruments) as described[39]. FLIC intensities were extracted using custom software written in LabView 2016 for Windows (National Instruments, Austin, TX) and fitted using FLIC v.0.5, which was generously provided by Armin Lambacher and Peter Fromherz (Max Planck Institute for Biochemistry, Martinsried/München, D 82152 Germany)[35]. The simulation of the fluorescent release lineshapes (Fig. 5c) was performed in Matlab_R2019a (Mathworks, Natick, MA). Fits of kinetic fusion data and assembly of data graphs

were performed using Igor Pro 8 (Wavemetrics, Lake Oswego, OR). EPR spectra and depth parameters were plotted using OriginPro, versions 7.5 or 2021 (OriginLab, Northhampton, MA). Molecular Structures were rendered using PyMol (Schrodinger, NY, NY).

**Reporting summary**. Further information on research design is available in the Nature Research Reporting Summary linked to this article.

## Data availability
Source data are provided with this paper. Any primary data not included in the Source Data file are available from either V.K. or D.S.C.

## Code availability
Code that was used for the processing of EPR data is available from D.S.C. upon request.

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

## Acknowledgements

This work was supported by NIH grant NIGMS GM072694 to L.K.T. and D.S.C. We would like to thank Dr. David A. Nyenhuis for providing software for the routine phasing and normalization of the EPR spectra.

## Author contributions

S.B.N. and N.K. designed the EPR experiments, expressed and labeled Syt1, performed power saturation and EPR measurements on Syt1 C2AB and analyzed the data. V.K. and A.K. designed and performed single-particle fusion assays on Syt1 C2AB and its mutants and analyzed the data. V.K. designed and performed FLIC data on SNARE proteins. B.L. expressed, isolated and reconstituted membrane associated SNAREs. A.T. performed membrane binding measurements on C2AB and carried out initial EPR power saturation measurements on C2AB mutants. L.K.T. and D.S.C. designed the experiments and analyzed the data. S.B.N., V.K., A.K., B.L., L.K.T., and D.S.C. wrote the manuscript.

## Competing interests

The authors declare no competing interests.
