## [Peer Review File · Nature Communications]

REVIEWER COMMENTS

Reviewer #1 (Remarks to the Author):

The manuscript by Nyenhuis et. al. focuses on the effect of two arginine residues in C2B of synaptotagmin 1 on vesicle fusion by controlling the expansion of the fusion pore. The authors address that the arginine apex positions the C2B domain at the curved membrane surface, i.e., hemifusion state, thereby leading to the expansion of fusion pore for neurotransmitter release. The authors apply EPR and single DCV fusion assay using chromaffin granules, also known as DCV, isolated from PC12 cells.

The authors present an interesting analysis using their well-established tools to monitor the efficiency of fusion in a single vesicle level and the interaction of syt-1 with SNARE and/or membrane. Their data presented in the manuscript are consistent with a novel model, suggesting that C2B domain interacts with the negatively curved membrane surface (probably hemifusion state) to mediate expansion of the fusion pore, but there are some points that need to be answered and updated for publication.

Major issues:

1. In Fig.2c, 285R1 shows calcium-dependency, but 350R1 doesn't. Explain and discuss why both have a different calcium dependency for membrane contact.
2. In Fig.3, the important control experiments are missing; i.e., PC:PS without calcium. It should be tested if RQRQ mutants disrupt membrane contact without calcium ions in order to address that the arginine apex is involved in membrane contact before vesicle fusion.
3. In Figure S1, the arginine apex of C2B contacts the membrane interface when membranes contain PIP2 instead of PS. The electrostatic force drives the interaction of the arginine apex with membrane and/or SNARE, so it make biological sense that the arginine apex can electrostatically interact with PIP2. Authors need to discuss possibility that the arginine apex might be able to bind to the plasma membrane as well as vesicle membrane. The effect of KAKA mutant on membrane contact should be compared with RQRQ mutant using PC:PIP2 membrane without calcium.
4. In Fig. 4, AAA mutation eliminates the Syt1C2AB-SNARE interaction, suggesting that Syt1 C2AB-SNARE interaction might be important for vesicle docking, but not essential for vesicle fusion, fusion pore opening and fusion kinetics. In Fig.5f, authors address that granule binding, i.e., docking, is mediated by SNAREs, but it should be explained in more details; it is mediated by Syt1-SNARE interaction or SNARE assembly.
5. In Fig. 5d legend, synaptotagmin isoforms, Syt1 and Syt9, were tested. But, Syt7 is one of major synaptotagmin isoforms in chromaffin granule (Ref.; Synaptotagmin-1 and -7 are functionally overlapping Ca²⁺ sensors for exocytosis in adrenal chromaffin cells. Schonn JS, Maximov A, Lao Y, Südhof TC, Sørensen JB, Proc Natl Acad Sci U S A. 2008 Mar 11; 105(10):3998-4003.). Syt7 should be depleted instead of Syt9.
6. Hemifusion is the possible intermediate where the apex contacts membrane in the absence of calcium during vesicle docking. Authors should explain and show some evidence of hemifusion, which might be extremely transient intermediate step.
7. The authors address that C2A domain interacts with vesicle membrane, not the plasma membrane. But, PS, anionic phospholipids, is present both in vesicle and plasma membrane and it is still possible that C2A domain binds to PS of the plasma membrane. The authors have to explain why C2A prefer vesicle membrane to interact rather than the plasma membrane.

Minor issues:

1. No lipid composition of PC:PS lipid vesicles was mentioned in Fig.2b,c and Fig.6a. Authors have to explain the lipid composition of PC:PS lipid vesicles in the method and figure legend. Company where phospholipid purchased and which types of PC and PS used should be described in the

method.

2. No composition of PC:PIP2 membrane was mentioned; 1% or 2% in Fig.3? Liposome consisting of only PC:PIP2 without any other lipids remains unstable. Authors need to make sure the structure of PC:PIP2 membrane.

Reviewer #2 (Remarks to the Author):

Referee report

Manuscript title: "Conserved Arginine Residues in Synaptotagmin 1 Regulate Fusion Pore Expansion Through Membrane Contact"

Authors: Sarah B. Nyenhuis et al.

Submitted to Nature Communications

The manuscript describes a detailed study designed to unravel the synaptic vesicle membrane pore formation, a crucial step in neurotransmission of signals. The role of synaptotagmin 1 is investigated by spin label EPR and membrane fusion and SNARE interactions are tested by auxiliary techniques. The manuscript presents a detailed and in depth study, combining spin label mutants with arginine mutations that change the behavior of synaptotagmin 1. The experiments are exhaustive, comprehensive and well designed. The authors tackle a multicomponent system, involving membrane interactions, SNARE and their protein of interest, any one of these separately would already be a publication to some!

The authors show that novel interaction are relevant and that the approach used is able to unravel them. The manuscript is comprehensible, clearly written and is warmly recommended for publication.

A few points that arouse when reading the manuscript for consideration of the authors:

1. While the authors describe the system very clearly in the introduction, the reader is missing an overview of which positions were used for the spin labelling and why they were chosen. It would be nice, if space permits, to add a paragraph that describes which positions will be discussed in the manuscript.
2. The authors use the MTSSL spin label that attaches to the protein with a disulfide bond. Isn't the reversibility of the bond a risk in the study? Apparently not, but I would be interested to hear what he considerations were. Consider this curiosity of the reviewer, not a question to be discussed in the manuscript per se.
3. The authors do not mention this explicitly and perhaps I overlooked it: What does PiP2 stand for and what is the precise charge of this membrane component.
4. In view of the careful phrasing of the manuscript, in the caption of Fig. 4 the "broadened component of the hyperfine tensor" is an unfortunate exception. It is not the hyperfine tensor that broadens, so that should be rephrased.

Reviewer #3 (Remarks to the Author):

This is an interesting study from Cafiso and colleagues, concerning the role of arginine resides, located on the face of the Syt1 C2B opposite the Ca²⁺ binding loops, in mediating protein-protein/lipid interactions important for exocytosis. The study utilizes a combination of EPR spectroscopy and TIRF-based imaging of individual fusion events to assess the role of the aforementioned portion of the C2B. The most striking data can be found in Figure 5 in which it is shown that mutations in the "arginine apex" of arginine to glutamine cause fusion phenotypes consistent with the complete absence of the C2AB. This result provides an explanation for a previous finding by Rizo and colleagues (2008), which showed RQ mutations at this region abrogated synchronous release in hippocampal neurons.

In general, the work here is well-done with text that is well-composed and relatively easy to follow. However, there are some confusing aspects to the writing which the authors should be able to resolve in a minor revision.

1. It is stated in In 405 in the discussion that "Syt7 demonstrates a substantially lower release rate than does WT Syt1 (23); however, the rates of release for Syt7 are similar to those seen here for either the RQ or RQRQ mutations of Syt1 (Figure 5f). A comparison of the sequence of the arginine apex for Syt1 and Syt7 shows that Syt7 is RQ rather than RR, and thus is identical in this regard to the Syt1 RQ mutant. We speculate that Syt7 and the RQ mutant of Syt1 look similar in terms of the rate of fusion pore opening because both proteins have the same residues in this apex."

However, no information on release 'rates' are provided in this study, which makes it difficult to square these results with what the authors previously published in Bendhamane et al. (2020). The del I does not report on rate as far as I can tell. Can the authors provide rate information for RQ, RQRQ, -C2AB, and +Syt7 C2AB (for comparison)?

2. Does the RQ mutation abolish release? Or does release still occur, but at a substantially slower rate? It is hard to know. If this mutation abolishes release, then of course these data would NOT be consistent with what the authors recently published using Syt7-labeled granules. It is also very confusing to me that the value of del I is more or less the same when the fusion assay is performed with granules lacking C2AB and with granules harboring Syt1 RQ mutants.

3. On In 232, the authors state that "The intensity of the spike in fluorescence (ΔIC) is indicative of the amount of content released from the granule during the fast phase of the fusion event after the initial slow release." This sentence is confusing to me and I wonder if it can be rephrased so that the actual point is made clearer. Do all events have a fast phase and slow phase? And does the fast phase always follow the slow phase? What is the time resolution of the experiments?

4. Just below the section referenced above, the authors state "The value of ΔIC reflects the mode of contents release, or the timing of the fusion event..." Can del I provide information on both "timing" and "mode"? If so, the authors should articulate their thinking on this. Presumably, a granule can fuse, release most of its contents quickly, and then be recovered at the site of fusion. In this case, the del I "spike" would be transient, yet the mode would reflect a non-canonical fusion mode (i.e., cavicapture).

5. A question related to what is asked in #3 -- Do all fusion events that are measured here result in complete release of fluorescent NPY-Ruby?

6. On In 258, the authors state, "However, when we examine the timing of the fusion event (the release mode)..." These terms – timing and release mode – are not interchangeable (see point #3 above). The timing of the fusion event might commonly be interpreted as the time at which a fusion event occurs with respect to some precipitating event (e.g., stimulation). Here, however, the authors seem to bestow upon it a different meaning, namely, release duration. The authors should clarify their wording in this section.

7. A minor correction – In the abstract, the authors use the word "probably" when they clearly meant to state "probability".

Response to Reviewers

(reviewer's comments/questions in italics)

Reviewer #1

The manuscript by Nyenhuis et. al. focuses on the effect of two arginine residues in C2B of synaptotagmin I on vesicle fusion by controlling the expansion of the fusion pore. The authors address that the arginine apex positions the C2B domain at the curved membrane surface, i.e., hemifusion state, thereby leading to the expansion of fusion pore for neurotransmitter release. The authors apply EPR and single DCV fusion assay using chromaffin granules, also known as DCV, isolated from PC12 cells.

The authors present an interesting analysis using their well-established tools to monitor the efficiency of fusion in a single vesicle level and the interaction of syt-1 with SNARE and/or membrane. Their data presented in the manuscript are consistent with a novel model, suggesting that C2B domain interacts with the negatively curved membrane surface (probably hemifusion state) to mediate expansion of the fusion pore, but there are some points that need to be answered and updated for publication.

Major issues:

1. In Fig.2c, 285R1 shows calcium-dependency, but 350R1 doesn't. Explain and discuss why both have a different calcium dependency for membrane contact.

Site 350R1 does show a Ca²⁺-dependency, but it is not as large as that seen at the other two sites, 285R1 and 349R1. In either case (with or without Ca²⁺), site 350R1 is close to the interface, an observation that is consistent with the fact that removing Ca²⁺ weakens but does not abolish membrane binding under the conditions of this experiment (see Supplementary Figure 4b). Since the C2B domain has different membrane bound orientations in the presence and absence of Ca²⁺ (Kuo et al, 2009, JMB, 387, 284), one would not expect each position to show the same difference in depth parameter with and without Ca²⁺

2. In Fig.3, the important control experiments are missing; i.e., PC:PS without calcium. It should be tested if RQRQ mutants disrupt membrane contact without calcium ions in order to address that the arginine apex is involved in membrane contact before vesicle fusion.

To simplify the presentation, we did not originally include these data, but a comparison of the depth data for both conditions (with and without Ca²⁺) is now provided in the Supporting Information (Supplementary Figure 2). We have also included the membrane binding data for the WT, RQ, and RQRQ mutant in the absence of Ca²⁺ (see Supplementary Figure 4b). These data are now referenced in a sentence on the last line of the first paragraph on page 7.

As seen in Figure S2, the RQ and RQRQ mutants disrupt the membrane interaction in the absence of Ca²⁺ as they do in the presence of Ca²⁺. The only difference is that for PC:PS bilayers, where the Ca²⁺ independent affinity is not as strong (see data Supplementary Figure 4b), the depth parameters are shifted to slightly more aqueous positions than in the presence of Ca²⁺.

3. In Figure S1, the arginine apex of C2B contacts the membrane interface when membranes contain PIP2 instead of PS. The electrostatic force drives the interaction of the arginine apex with membrane and/or SNARE, so it make biological sense that the arginine apex can electrostatically interact with PIP2. Authors

need to discuss possibility that the arginine apex might be able to bind to the plasma membrane as well as vesicle membrane. The effect of KAKA mutant on membrane contact should be compared with RQRQ mutant using PC:PIP2 membrane without calcium.

We have now included data comparing the membrane binding of the RQRQ and KAKA mutations for Syt1 C2AB w/o Ca²⁺ in the presence of PC:PIP2, PC:PS and PC:PS:PIP2 (Supplementary Figure 4b). The KAKA mutant fails to bind without Ca²⁺ unless both PS and PIP2 are present. In contrast, the RQRQ mutant still binds with virtually the same affinity as the WT even in the absence of Ca²⁺.

In the absence of Ca²⁺, the arginine apex will bind to the plasma membrane rather than the vesicle membrane. PIP2 is primarily in the plasma membrane, and in the absence of Ca²⁺, the affinity of C2B will be much higher to the bilayer containing PIP2 than to the vesicle membrane (see Pérez-Lara, et al. (2016) eLife). Because of PIP2, we expect the apex to contact the plasma membrane at low Ca²⁺ levels. Increasing Ca²⁺ should trigger C2B to insert into the nearest membrane (the plasma membrane). As the vesicle and plasma membrane merge, we imagine (as depicted in Figure 7) that C2B may orient to the curved surface formed by a developing fusion pore. We do not think the apex will interact with the vesicle membrane prior to Ca²⁺ levels rising. We added a sentence in the legend to Figure 7 to clarify this point.

4. In Fig. 4, AAA mutation eliminates the Syt1C2AB-SNARE interaction, suggesting that Syt1 C2AB-SNARE interaction might be important for vesicle docking, but not essential for vesicle fusion, fusion pore opening and fusion kinetics. In Fig.5f, authors address that granule binding, i.e., docking, is mediated by SNAREs, but it should be explained in more details; it is mediated by Syt1-SNARE interaction or SNARE assembly.

In this assay, previous work has demonstrated that docking in the absence of Ca²⁺ is dependent on the formation of the SNARE complex (Kreutzberger et al. Sci Adv. 2017). Docking is not observed in the absence of either t-SNAREs or in the presence of a synaptobrevin peptide inhibitor. The knock down of synaptotagmin does not affect docking (Kreutzberger et al Sci Adv. 2017) and the addition of the soluble C2AB domain does not enhance or inhibit docking (Kiessling et al. NSMB 2018). We changed the wording in the second paragraph on page 10 to make these points clearer.

5. In Fig. 5d legend, synaptotagmin isoforms, Syt1 and Syt9, were tested. But, Syt7 is one of major synaptotagmin isoforms in chromaffin granule (Ref.; Synaptotagmin-1 and -7 are functionally overlapping Ca²⁺ sensors for exocytosis in adrenal chromaffin cells. Schonn JS, Maximov A, Lao Y, Südhof TC, Sørensen JB, Proc Natl Acad Sci U S A. 2008 Mar 11; 105(10):3998-4003.). Syt7 should be depleted instead of Syt9.

On this point, we disagree with the reviewer. The paper that the reviewer refers to uses primary mouse adrenal cells to study synaptotagmin isoforms on granules, while the purification we use here is from an immortalized PC12 cell line. The granule purification we use has been carefully characterized, as described previously (Kreutzberger et al. 2017 Sci Adv; Kiessling et al. 2018 NSMB; and Kreutzberger et al. 2019 Nat Comm), and in two recent papers we examined the synaptotagmin isoforms that are present on these purified granules (Kreutzberger et al. Nat Comm 2019 and Bendhamane et al. 2020 J NeuroChem). An examination of the Western blots in Kreutzberger et al. 2019 Nat Comm (Supplementary Figure 1) shows that the predominant synaptotagmin isoforms in the secretory vesicles obtained from PC12 cells are synaptotagmin-1 and -9 (see fraction 9, which is the

fraction used for DCVs) while synaptotagmin-7 is absent. We have also demonstrated that we can express synaptotagmin isoforms (such as synaptotagmin-7) within the background of this synaptotagmin-1 and -9 knockdown (Kreutzberger et al. Nat. Comm. 2019 and Bendhamane et al 2020 J NeuroChem). This work was referenced in the discussion and was used to characterize the release behavior of synaptotagmin-7 granules.

6. Hemifusion is the possible intermediate where the apex contacts membrane in the absence of calcium during vesicle docking. Authors should explain and show some evidence of hemifusion, which might be extremely transient intermediate step.

We are using the release of a labeled peptide to read out fusion of the DCVs with the supported bilayer. This type of read out reports only productive full fusion events and does not have the ability to show hemifusion intermediates. We have on-going work using membrane labels – but have seen no clear productive hemifusion intermediates within the time resolution of our experiments.

7. The authors address that C2A domain interacts with vesicle membrane, not the plasma membrane. But, PS, anionic phospholipids, is present both in vesicle and plasma membrane and it is still possible that C2A domain binds to PS of the plasma membrane. The authors have to explain why C2A prefer vesicle membrane to interact rather than the plasma membrane.

We agree with the reviewer that there is unlikely to be a strict preference for C2A to bind only to the vesicle membrane. Our earlier pulse EPR work on full-length Syt1 indicated that the binding of C2A to the vesicle membrane and C2B to the plasma membrane occurred and was likely the favored orientation (Nyenhuis et al., Biophys J 2019). Hence, we gave C2AB the orientation shown in Figure 7. However, we cannot exclude the possibility that some population of C2A interacts with the plasma membrane. To make this point clear, we added a note on the orientation in the legend for Figure 7.

Minor issues:

1. No lipid composition of PC:PS lipid vesicles was mentioned in Fig.2b,c and Fig.6a. Authors have to explain the lipid composition of PC:PS lipid vesicles in the method and figure legend. Company where phospholipid purchased and which types of PC and PS used should be described in the method.

The lipid compositions were mentioned in methods section, but to be clear we have included this information in the legends to Figures 2, Figure 6 and Supplementary Figure 1. We added information on the supplier to the methods section.

2. No composition of PC:PIP2 membrane was mentioned; 1% or 2% in Fig.3? Liposome consisting of only PC:PIP2 without any other lipids remains unstable. Authors need to make sure the structure of PC:PIP2 membrane.

We have included the composition into the legend in Figure 3. The reviewer is correct, liposomes with PC:PIP2 are unstable. We found that we had to use freshly prepared lipid vesicles with PIP2 for each experiment. If we did not, the vesicles degraded, and our results were not consistent. This is now indicated in Methods.

Reviewer #2

The manuscript describes a detailed study designed to unravel the synaptic vesicle membrane pore formation, a crucial step in neurotransmission of signals. The role of synaptotagmin 1 is investigated by spin label EPR and membrane fusion and SNARE interactions are tested by auxiliary techniques. The manuscript presents a detailed and in depth study, combining spin label mutants with arginine mutations that change the behavior of synaptotagmin 1. The experiments are exhaustive, comprehensive and well designed. The authors tackle a multicomponent system, involving membrane interactions, SNARE and their protein of interest, any one of these separately would already be a publication to some!

The authors show that novel interaction are relevant and that the approach used is able to unravel them. The manuscript is comprehensible, clearly written and is warmly recommended for publication.

A few points that arouse when reading the manuscript for consideration of the authors:

1. While the authors describe the system very clearly in the introduction, the reader is missing an overview of which positions were used for the spin labelling and why they were chosen. It would be nice, if space permits, to add a paragraph that describes which positions will be discussed in the manuscript.

The sites were selected in the C2B apex to be outward facing at the ends of turns on the C2 domain and we avoided replacing positively charged residues with the label. The label rotamers were also examined to be sure that they did not interfere with any critical side chains (such as the conserved arginine residues in the apex). Several sites in the apex were found to inhibit proper folding and function of the domain and we avoided these sites. In the Ca²⁺ binding region and polybasic face, sites were also outward facing and were chosen because they did not (based upon previous work) interfere with protein function. We added a few words in the introduction and several sentences in the first paragraph in the Methods section to briefly describe the basis for their selection.

2. The authors use the MTSSL spin label that attaches to the protein with a disulfide bond. Isn't the reversibility of the bond a risk in the study? Apparently not, but I would be interested to hear what he considerations were. Consider this curiosity of the reviewer, not a question to be discussed in the manuscript per se.

As the reviewer notes, the MTSSL label produces a labeled sidechain that is linked through a disulfide, and therefore reversible in an appropriate reducing environment. But, unless one has DTT or some other reagent present, the label is very stable and does not spontaneously reverse. It has been the preferred label for study for well over 25 years several reasons. First, it is highly reactive and specific towards Cys, even more so than reagents that produce a more stable C-S bond (iodoacetamide or maleimide labels for example). Second, we have many crystal structures of the label in globular and membrane proteins and we understand what determines its EPR lineshapes and what the available rotamers are for this sidechain. Finally, the label does not sample as large a space

as do many other labels, and it therefore produces distance distributions using methods such as DEER that are more tightly coupled to the protein conformation.

3. The authors do not mention this explicitly and perhaps I overlooked it: What does PiP2 stand for and what is the precise charge of this membrane component.

We have defined PIP2 in the introduction and given an “approximate” value for its valence as -4. Unfortunately, the valence of this lipid is not precisely defined. The valence can range from -3 to -5 depending upon pH, ionic conditions, and whether the lipid is associated with a protein. A more detailed discussion of its valence and properties can be found in an older review article from Stuart McLaughlin’s group (Ann Rev Biophys Biomol Struct 31, (2002) 151-175), which we have referenced.

4. In view of the careful phrasing of the manuscript, in the caption of Fig. 4 the “broadened component of the hyperfine tensor” is an unfortunate exception. It is not the hyperfine tensor that broadens, so that should be rephrased.

The reviewer is correct. We have re-worded the sentence in the legend to Figure 4.

Reviewer #3

This is an interesting study from Cafiso and colleagues, concerning the role of arginine residues, located on the face of the Syt1 C2B opposite the Ca²⁺ binding loops, in mediating protein-protein/lipid interactions important for exocytosis. The study utilizes a combination of EPR spectroscopy and TIRF-based imaging of individual fusion events to assess the role of the aforementioned portion of the C2B. The most striking data can be found in Figure 5 in which it is shown that mutations in the “arginine apex” of arginine to glutamine cause fusion phenotypes consistent with the complete absence of the C2AB. This result provides an explanation for a previous finding by Rizo and colleagues (2008), which showed RQ mutations at this region abrogated synchronous release in hippocampal neurons.

In general, the work here is well-done with text that is well-composed and relatively easy to follow. However, there are some confusing aspects to the writing which the authors should be able to resolve in a minor revision.

1. It is stated in ln 405 in the discussion that “Syt7 demonstrates a substantially lower release rate than does WT Syt1 (23); however, the rates of release for Syt7 are similar to those seen here for either the RQ or RQRQ mutations of Syt1 (Figure 5f). A comparison of the sequence of the arginine apex for Syt1 and Syt7 shows that Syt7 is RQ rather than RR, and thus is identical in this regard to the Syt1 RQ mutant. We speculate that Syt7 and the RQ mutant of Syt1 look similar in terms of the rate of fusion pore opening because both proteins have the same residues in this apex.”

However, no information on release ‘rates’ are provided in this study, which makes it difficult to square these results with what the authors previously published in Bendhamane et al. (2020). The del I does not report on rate as far as I can tell. Can the authors provide rate information for RQ, RQRQ, -C2AB, and

+Syt7 C2AB (for comparison)?

The reviewer is correct, we did not use the correct terminology. Our data for contents release, ΔI_c , does not show release rates, but instead reflects the release mode or the timing of the fusion event. At the single vesicle level, there are two components or steps to the release as seen in Figure 5c. In step 1, the content is released slowly (probably through a narrow pore) followed by step 2 that involves pore expansion. In step 2, the remaining content is released within one resolved time step. The second step leads to the observed ΔI_c which depends on how much content was left in the vesicle at that point. If the first step takes longer (slow pore expansion, blue curve Figure 5c), less content is left for the second step and smaller values for ΔI_c are observed compared to the case when the first step is short (fast pore expansion, red curve Figure 5c).

We changed the wording in this paragraph (5th paragraph in the Discussion) to accurately reflect what was compared in these two studies between the Syt1 apex mutations and the behavior of the Syt7 isoform.

2. Does the RQ mutation abolish release? Or does release still occur, but at a substantially slower rate? It is hard to know. If this mutation abolishes release, then of course these data would NOT be consistent with what the authors recently published using Syt7-labeled granules. It is also very confusing to me that the value of ΔI is more or less the same when the fusion assay is performed with granules lacking C2AB and with granules harboring Syt1 RQ mutants.

The RQ and RQRQ mutations do not abolish release, they simply fail to accelerate the timing of the fusion event which normally takes place with WT Syt1, Ca²⁺ and PIP₂. It should be noted that the release modes observed with no C2AB or with RQ mutants (or with Syt7) all look the same, i.e. longer duration of step 1 and therefore less content release (smaller ΔI) during step 2. So, yes, the RQ mutant still initiates release but does not change the release mode compared to the mode observed in the absence of Ca²⁺ or in the absence of C2AB altogether.

To clarify this point, we added a sentence at the end of the 5th paragraph in the Discussion

3. On ln 232, the authors state that “The intensity of the spike in fluorescence (ΔI_C) is indicative of the amount of content released from the granule during the fast phase of the fusion event after the initial slow release.” This sentence is confusing to me and I wonder if it can be rephrased so that the actual point is made clearer. Do all events have a fast phase and slow phase? And does the fast phase always follow the slow phase? What is the time resolution of the experiments?

To address this question, and we re-worded the sentence in this paragraph (last paragraph, page 9 going onto page 10) and re-wrote several sentences in the previous paragraph to clarify the meaning of this fluorescence signal. The time resolution in these experiments is 200 ms. Under these conditions the fast release phase always follows the slow release phase. It is possible to eliminate the first phase (at least within the time resolution of the experiment) by changing the asymmetry of the membrane (see Kreutzberger et al. (2017) Biophys J).

4. Just below the section referenced above, the authors state “The value of ΔIC reflects the mode of contents release, or the timing of the fusion event...” Can del I provide information on both “timing” and “mode”? If so, the authors should articulate their thinking on this. Presumably, a granule can fuse, release most of its contents quickly, and then be recovered at the site of fusion. In this case, the del I “spike” would be transient, yet the mode would reflect a non-canonical fusion mode (i.e., cavicapture).

See the answer to point 1 above for an explanation of mode. Cavicapture events would be hard to detect with content dye if they are faster than 200ms. To study cavicapture we would need to change the assay to include a membrane label and perhaps examine Ca²⁺ triggered fusion that includes Munc18 and Cpx.

5. A question related to what is asked in #3 -- Do all fusion events that are measured here result in complete release of fluorescent NPY-Ruby?

We typically observed a full release of vesicle contents. In those few cases where some residual fluorescence remains after the decay, we are not able to distinguish between content left inside the vesicle or content trapped at the site of fusion.

6. On ln 258, the authors state, “However, when we examine the timing of the fusion event (the release mode)...” These terms – timing and release mode – are not interchangeable (see point #3 above). The timing of the fusion event might commonly be interpreted as the time at which a fusion event occurs with respect to some precipitating event (e.g., stimulation). Here, however, the authors seem to bestow upon it a different meaning, namely, release duration. The authors should clarify their wording in this section.

We changed the wording in the paragraph (end of page 9 to 10) to clarify how we are defining fusion timing.

7. A minor correction – In the abstract, the authors use the word “probably” when they clearly meant to state “probability”.

We have corrected the typo.

REVIEWERS' COMMENTS

Reviewer #1 (Remarks to the Author):

Authors answered and addressed major comments and this revised article is well updated for publication. I am pleased to confirm the acceptance of the article.

Reviewer #3 (Remarks to the Author):

The authors have revised the text to my satisfaction.